# Structures of ISCth4 transpososomes reveal the role of asymmetry in copy-out/paste-in DNA transposition

Dalibor Kosek [ID], Alison B Hickman, Rodolfo Ghirlando, Susu He[†] & Fred Dyda[*] [ID]

## Abstract

Copy-out/paste-in transposition is a major bacterial DNA mobility pathway. It contributes significantly to the emergence of antibiotic resistance, often by upregulating expression of downstream genes upon integration. Unlike other transposition pathways, it requires both asymmetric and symmetric strand transfer steps. Here, we report the first structural study of a copy-out/paste-in transposase and demonstrate its ability to catalyze all pathway steps *in vitro*. X-ray structures of ISCth4 transposase, a member of the IS256 family of insertion sequences, bound to DNA substrates corresponding to three sequential steps in the reaction reveal an unusual asymmetric dimeric transpososome. During transposition, an array of N-terminal domains binds a single transposon end while the catalytic domain moves to accommodate the varying substrates. These conformational changes control the path of DNA flanking the transposon end and the generation of DNA-binding sites. Our results explain the asymmetric outcome of the initial strand transfer and show how DNA binding is modulated by the asymmetric transposase to allow the capture of a second transposon end and to integrate a circular intermediate.

**Keywords** antibiotic resistance; crystallography; mechanism; promoter; transposon

**Subject Categories** DNA Replication, Recombination & Repair; Microbiology, Virology & Host Pathogen Interaction; Structural Biology

The EMBO Journal (2021) 40: e105666

## Introduction

Transposons (or transposable elements, TE) are mobile genetic elements found in all living organisms and are important evolutionary shaping forces (Biémont & Vieira, 2006). The simplest prokaryotic autonomous DNA TEs are the insertion sequences (IS). They are usually formed by one or two open reading frames (ORFs) encoding a transposase enzyme (Tnp) essential for mobility and two terminal sequences typically arranged as terminal inverted repeats (TIRs) of similar but not necessarily identical sequence (Mahillon & Chandler, 1998). In bacteria, ISs have been linked to the emergence and dissemination of antibiotic resistance due in part to their ability to mobilize other genes in the form of composite TEs (Partridge *et al*, 2018). Many ISs also contain sequences that can act as promoters for genes located outside of the element and thus can dynamically affect their expression upon integration (Nevers & Saedler, 1977; reviewed in Siguier *et al*, 2015; Vandecraen *et al*, 2017; Babakhani & Oloomi, 2018). These properties of ISs have contributed to the rise of multidrug-resistant bacterial strains defined as urgent threats by the Centers for Disease Control and Prevention (Alekshun & Levy, 2007; McKenna, 2013; Watkins & Bonomo, 2016; CDC 2019) such as antibiotic-resistant *C. difficile*, and carbapenem-resistant *Enterobacteriaceae* and *N. gonorrhoeae*. Understanding the molecular mechanisms of mobilization and upregulation of antibiotic resistance is imperative for the development of new approaches to control the emergence of resistant strains. Such understanding may also provide an opportunity to develop novel gene delivery and modification tools for use in research as well as human medicine (Haapa *et al*, 1999; Izsvák & Ivics, 2004; Adey *et al*, 2010; Sakanaka *et al*, 2018).

TEs can be mobilized in a number of different ways. In eukaryotes, DNA transposition is largely carried out by cut-and-paste TEs (Fig 1A), whereas in bacteria replicative DNA transposition pathways dominate (Siguier *et al*, 2015). In cut-and-paste DNA transposition, double-stranded breaks are made by the Tnp at both ends of the TE close to the TIRs which serve as the Tnp-binding sites. The liberated TE leaves behind an empty donor site. Subsequently, the TE integrates into a target site. During replicative DNA transposition (Fig 1B and C), the Tnp nicks only a single strand at the TE end and transposition intermediates needed for mobility are then generated by the host DNA replication/repair machinery. In all replicative pathways, the original TE copy is regenerated at the donor site and a new copy emerges at the integration site. Therefore, a replicative transposition event always increases the copy number of the TE.

The well-studied "cointegrative" replicative DNA transposition pathway (Fig 1B) goes through a "Shapiro intermediate", followed by DNA replication that generates a cointegrate with two TE copies;

Laboratory of Molecular Biology, National Institute of Diabetes and Digestive and Kidney Diseases, National Institutes of Health, Bethesda, MD, USA
*Corresponding author. Tel: +1 301 402 4496; E-mail: Fred.Dyda@nih.gov
†Present Address: State Key Laboratory of Pharmaceutical Biotechnology, Medical School of Nanjing University, Nanjing, Jiangsu, China

The EMBO Journal   40: e105666 | 2021   **1 of 22**

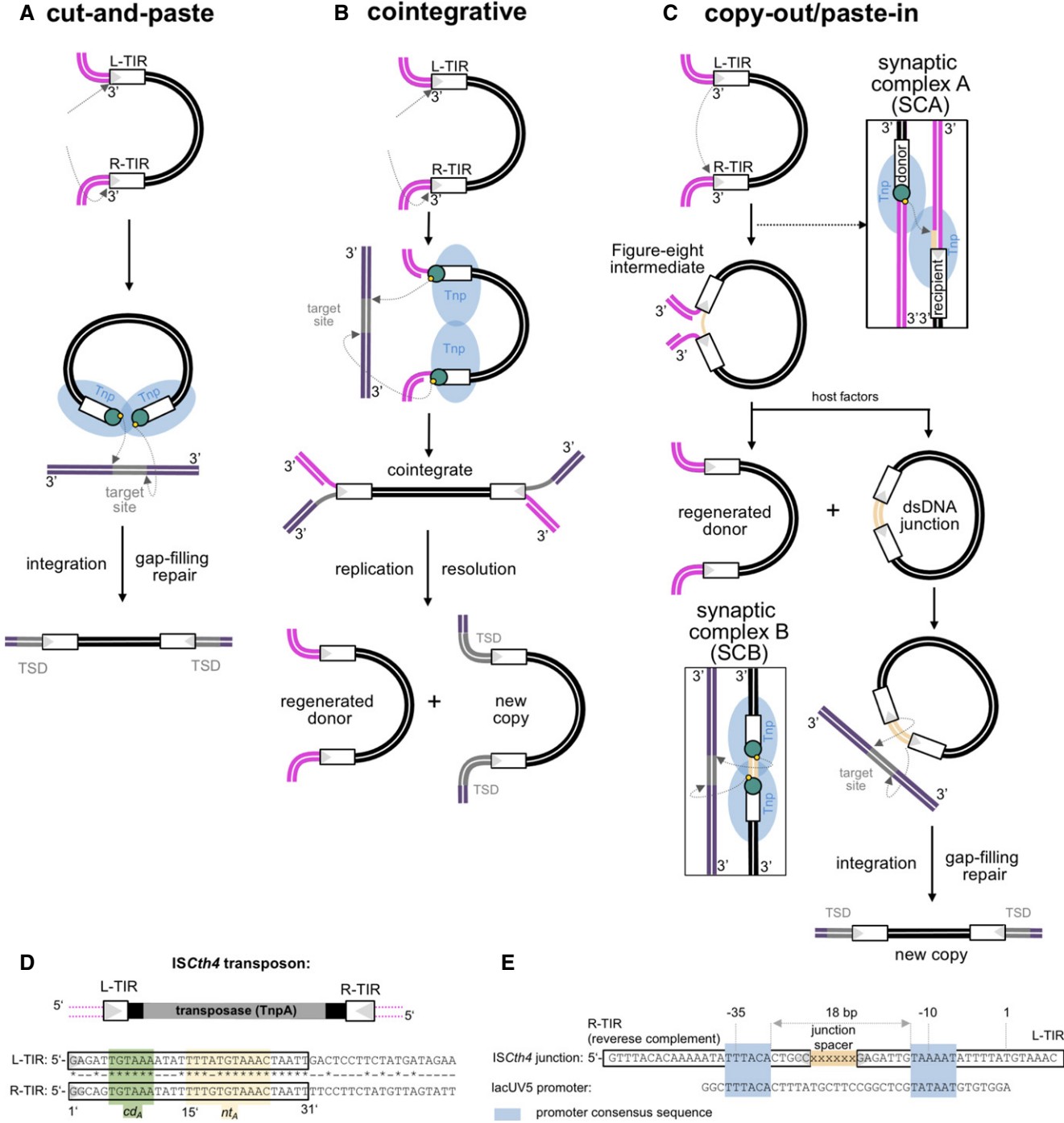

**Figure 1.   Schematics of common prokaryotic transposition pathways and ISCth4 transposon.**

A–C   Schematics of transposition pathways. Transposon DNA is in black, flanking DNA in magenta, TIRs as white boxes, target site in gray, and sequences flanking the target site in violet, target site duplications are labeled as TSD. Tnp, transposase enzyme (shown schematically as a single blue oval). Green circles represent transposase active site bound at the ends of transferred TIRs. Arrows indicate cleavage or strand transfer reactions occurring at TIR ends. Light orange indicates the initially single-stranded spacer derived from attack on flanking DNA that is subsequently converted to double-stranded form. The yellow dots indicate nucleophilic 3′-OH groups. Resolution of cointegrate intermediates has been described previously (Chaconas & Harshey, 2002). 3′ ends of DNA are indicated. In (C), only transfer from L-TIR to R-TIR is depicted for simplicity.

D   Schematic of the ISCth4 transposon (top) and aligned sequences of its L-TIR and R-TIR (bottom). Nucleotides identical on both ends are indicated with asterisks. Sequences bound by $cd_A$ and $nt_A$ sites, established herein, are indicated.

E   Alignment of an ISCth4 dsDNA junction intermediate with a 6-bp junction (in light orange) and the lacUV5 promoter (Fuller, 1982). The −35 and −10 sequences are highlighted in blue boxes.

this is subsequently resolved to regenerate the donor DNA and a target DNA containing the integrated TE (Shapiro, 1979; Chaconas & Harshey, 2002). A representative transpososome assembly has been visualized using X-ray crystallography (Montaño *et al*, 2012). In contrast, the widespread replicative "copy-out/paste-in" pathway used by a majority of bacterial IS families (Chandler *et al*, 2015; Siguier *et al*, 2015) is characterized by a distinct branched DNA intermediate, referred to as the "Figure-eight" intermediate (Fig 1C). The Figure-eight intermediate is formed by single-stranded cleavage at the end of one of the TIRs (so-called "donor") that generates a free 3′-OH which subsequently attacks the flanking sequence just adjacent to the second TIR ("recipient") (Fig 1C, Polard & Chandler, 1995; Polard *et al*, 1992; Lewis & Grindley, 1997; Sekine *et al*, 1999). The Figure-eight intermediate thus contains a TIR-TIR junction in which the two TIRs are separated by a single-stranded spacer of a few nucleotides. This asymmetric intrastrand transfer step is the unique feature of the copy-out/paste-in pathway because in other dsDNA transposition pathways, reactions at the two ends of TE are always identical (Fig 1A and B). This is reflected in the rotational symmetry relating the two bound TIRs seen in available transpososome structures (Davies *et al*, 2000; Richardson *et al*, 2009; Montaño *et al*, 2012; Liu *et al*, 2019).

The protein–DNA complex that catalyzes the asymmetric intrastrand transfer step and generates the Figure-eight intermediate has been referred to as synaptic complex A (SCA, Fig 1C). DNA replication converts the Figure-eight intermediate into a fully dsDNA circular intermediate (Duval-Valentin *et al*, 2004; Loot *et al*, 2004) that is subsequently integrated at the target site. In the intermediate, the two TIRs and the spacer separating them often form a fusion promoter as initially observed for tandemly inserted ISs (Dalrymple, 1987; Reimmann *et al*, 1989) which drives increased transposase expression (Prentki *et al*, 1986; Duval-Valentin *et al*, 2001). This is thought to be a crucial point of regulation for copy-out/paste-in TEs so that high levels of the transposase are generated only when a suitable intermediate is present. The promoter forms as the −35 and −10 binding sites of the sigma factor of the RNA polymerase holoenzyme are divided between the two TIRs, and the proper spacing between them is ensured by the length of the spacer (Szeverényi *et al*, 1996; Lewis & Grindley, 1997; Ton-Hoang *et al*, 1997).

A different protein–DNA complex, synaptic complex B (SCB; Fig 1C), is proposed to bind the dsDNA circular intermediate, nick the 3′-ends of the TIRs, and integrate it symmetrically into target DNA. After DNA repair, characteristic target site duplications (TSD) are generated flanking the IS at its new genomic location (Rousseau *et al*, 2008). One important consequence of TIRs that contain one half of a fusion promoter is that their integration can affect transcription patterns of genes near the integration site. Indeed, the integration of copy-out/paste-in elements just upstream of antibiotic resistance genes appears to be especially important in the emergence of antibiotic resistances (Kamruzzaman *et al*, 2015; Vandecraen *et al*, 2017).

Despite its importance in the spread of antibiotic resistance, our understanding of the molecular mechanism leading to copy-out/paste-in transposition is limited by the absence of relevant structural data. Analysis of the IS*3*, IS*256*, and IS*30* families (shown to transpose via copy-out/paste-in) revealed that their TIRs contain bipartite transposase-binding sites, one close to the TE end (terminal site)

which is bound by the C-terminal catalytic domain of the Tnp and a subterminal site bound by an N-terminal DNA-binding domain (Normand *et al*, 2001; Nagy *et al*, 2004; Hennig & Ziebuhr, 2010; Lewis *et al*, 2011). Here, we report the first structural results of a Tnp that is able to catalyze all the steps of the copy-out/paste-in pathway *in vitro* bound to DNA substrates reflecting three different steps in the pathway: the pre-reaction state with flanking DNA, the pre-cleaved state, and the product of strand transfer from one transposon end into the recipient end. Together, the structures suggest a molecular mechanism of asymmetric and symmetric strand transfer reactions that are characteristic of the copy-out/paste-in DNA transposition pathway.

# Results

## Identification of IS*Cth4* as an experimental model

To date, only a handful of copy-out/paste-in TEs have been studied in detail and there is no mechanistic information on any element from this group that might provide insight into their unique strand transfer reactions (reviewed in Siguier *et al*, 2015; Chandler *et al*, 2015). The main obstacles have been poor solubility or stability of the relevant proteins and complexes (Rousseau *et al*, 2010; Lewis *et al*, 2012). In an attempt to identify a system that might be amenable to *in vitro* characterization, we screened several Tnps from extremophilic bacteria for expression in *Escherichia coli* and solubility. Among them, the 47 kDa transposase of IS*Cth4* (a 1,503-bp IS in *Clostridium thermocellum,* strain ATCC 27405) was stable and readily purifiable (Fig 1D and Appendix Fig S1A).

IS*Cth4* belongs to the IS*256* family of ISs (Siguier *et al*, 2006; Kichenaradja *et al*, 2010), whose eponymous IS carries out reactions consistent with copy-out/paste-in transposition and which forms dsDNA circular IS*256* species *in vivo* (Loessner *et al*, 2002; Prudhomme *et al*, 2002; Hennig & Ziebuhr, 2010). Tnps of the IS*256* family have an RNase H-like catalytic domain and are similar to Tnps of the eukaryotic *Mutator*-like transposable elements superfamily (MULEs, Eisen *et al*, 1994; Hua-Van & Capy, 2008). A single ORF encodes the IS*Cth4* transposase (hereafter, TnpA). Primary sequence alignment with the IS*256* Tnp (with which it shares 26% sequence identity; Appendix Fig S2) indicates that the hallmark DDE active site residues are D175, D241, and E348 (Haren *et al*, 1999). TnpA also has a predicted α-helical insertion domain within the RNase H-like catalytic domain that carries a C/DxxH motif (C262/H265). This motif is important for catalysis among Tnps with an α-helical insertion domain (Liu & Wessler, 2017; Hickman *et al*, 2018). Twenty-four of the 31 bp of the left TIR (L-TIR; sometimes referred to as IRL) and the right TIR (R-TIR, or IRR) are almost identical, and most of the variation is within the first 14 bp (Fig 1D). Each TIR contains a subterminal motif, 5′-TGTAAA-3′, previously noted as a conserved feature of IS*256* family members (Dodd *et al*, 1994). There are 15 copies of IS*Cth4* in its host genome, most of which are flanked by 8-bp TSDs, strongly suggesting that it is active *in vivo* (Data ref: Copeland *et al*, 2011).

There are sequences closely related to −35 (TTTACA) and −10 (TAAAAT) promoter elements within the R- and L-TIR, respectively (Fig 1E), that would assemble a functional promoter if they were joined head-to-head with a 5- or 6-bp spacer, as demonstrated for

IS*256* hybrid promoters (Maki & Murakami, 1997). These promoter elements resemble those shown to be functional in *C. thermocellum* (Olson *et al*, 2015). Thus, a key lifestyle feature of copy-out/paste-in elements appears to be retained in IS*Cth4*.

## TnpA catalyzes all the steps of copy-out/paste-in transposition

To verify that IS*Cth4* is active, we reconstituted the two characteristic strand transfer reactions of the copy-out/paste-in pathway *in vitro*. Although we were unable to directly detect the strand transfer step that generates the Figure-eight intermediate using oligonucleotides, we recapitulated Figure-eight intermediate formation using a PCR-based *in vitro* strand transfer plasmid assay in which we introduced the 35-bp L-TIR and R-TIR sequences into pUC19 ("pUC19LR"; Fig 2A) or the L-TIR alone ("pUC19L"). When the substrate pUC19LR was supercoiled (Fig 2B, lanes 3, 4) or linearized pUC19L was used (Fig 2B, lane 1), the expected ~ 400-bp PCR product corresponding to a TIR–TIR junction was not detected; however, incubation of TnpA with linearized pUC19LR yielded a robust PCR signal (Fig 2B, lane 2). Sequencing of the PCR product identified four different targeted strand transfer products (out of eight recovered) corresponding to junctions with spacer sequence varying in length from 6 to 8 bp (Fig 2C, top). The spacer sequences are readouts of strand transfer direction, and in five cases, the R-TIR was the donor joined to the flanking sequence of the L-TIR recipient while the opposite was true in the remaining three cases. This suggests that TnpA can use either TIR to strand transfer into the other. These results are consistent with those previously reported for IS*256* which generated circular dsDNA intermediates with varying spacer sizes (5, 6, 7, and 19 bp), formed by attack of either end on the other TIR (Loessner *et al*, 2002; Prudhomme *et al*, 2002).

The second characteristic strand transfer reaction is the integration of the dsDNA circular intermediate. This is initiated by single-stranded cleavage at each TIR to generate the nucleophilic 3′-OH groups needed for symmetric strand transfer into a target (Fig 1C). We first asked if pre-cleaved TIRs with free 3′-OHs could be integrated into a supercoiled plasmid. In this assay, integration of one TIR results in a relaxed plasmid whereas concerted integration of two TIRs yields a linearized plasmid (Fig 2D). TnpA readily integrated R-TIRs longer than 25 bp into pUC19 and generated a linear reaction product, whereas R-TIRs shorter than 25 bp had only low integration activity as indicated by faint linear reaction products (Fig 2E, lanes 4–12 vs. lanes 13–27). We did not detect integration products when we used a 26-mer with a sequence unrelated to the TIRs (Fig 2F, lanes 3–5) or an oligonucleotide in which the first 14 bp of the R-TIR were replaced by random sequence (r14R12, Fig 2F, lanes 6–8).

We then asked if coupled cleavage and integration into a pUC19 plasmid could be detected using ~ 80-bp oligonucleotides with two TIRs separated by a spacer between mimicking the circular dsDNA intermediate (R35(j*x*)L35 where *x* refers to the length of the spacer; Fig 2G and H). We varied the length of the spacer from 0 to 10 bp and observed that a linear product was formed consistent with integration of two TIR ends (Fig 2G). The reaction was the most efficient when the TIRs were separated by a 6-bp spacer (Fig 2G, lanes 15–17) although linear products were detectable with all spacer lengths tested. To confirm that integration had occurred, the linear product of the reaction with the 6-bp spacer junction was cloned

and sequenced, and we recovered five integration events (Fig 2I, top). In all cases, the TSD was 8 bp long and there appeared to be a preference for integration into A/T-rich sequences. This is consistent with the A/T-rich 8-bp TSDs observed in the genome of *C. thermocellum* and for IS*256* itself (Loessner *et al*, 2002; Kleinert *et al*, 2017). Neither TIR junction integration (Appendix Fig S1B) nor Figure-eight formation (Appendix Fig S1C) was detected using TnpA with the active site mutation D175A. Although *C. thermocellum* is an anaerobic thermophile that grows optimally at 55°C (Akinosho *et al*, 2014), we did not observe any effect on *in vitro* activity when the temperature was increased.

## Both TIRs from a TIR junction are integrated with a preference for supercoiled target DNA

To further investigate *in vitro* junction integration, we carried out the reaction with both TIR termini of the junction substrate (R35(j6)L35) 5′-fluorescently labeled with FAM or Cy5 (shown schematically in Fig 3A). We also tested a junction pre-nicked at the L-TIR (R35(j6)L35nick-3′OH) and a pre-nicked junction in which the nucleotide on L-TIR that would typically provide the nucleophilic 3′-OH was replaced by a dideoxynucleotide (R35(j6)L35nick-ddC). Both R35(j6)L35 (Fig 3A, lanes 7–9) and R35(j6)L35nick-3′OH (Fig 3A, lanes 16–18) yielded the same reaction products with both TIRs integrated into the linearized plasmid product, consistent with coupled cleavage and integration. In contrast, R35(j6)L35nick-ddC (Fig 3A, lanes 13–15) produced only a relaxed target plasmid containing the 5′-FAM label, indicating that only the R-TIR had been integrated. The absence of linearized target plasmid in lanes 13–15 indicates that when only one end of a junction can be integrated, TnpA does not utilize another junction substrate from the reaction mixture. This suggests that the linear products observed for R35(j6)L35 and R35(j6)L35nick-3′OH and their parallel 5′-FAM and 5′-Cy5 signals were due to concerted integration from a single junction substrate. As a control, we carried out the integration reaction with labeled R35r41, an oligonucleotide where L-TIR was replaced with non-related DNA; the result shows only poor integration of R-TIR in comparison with R35(j6)L35 (Fig 3A, lanes 4–6 vs. lanes 7–9). The predominant product with R35r41 is the linearized target plasmid labeled only with 5′-FAM, indicating that—in contrast to a compromised junction—when a TIR end with random flanking DNA is used as a substrate, two DNA molecules are integrated.

We also asked if the product of the asymmetric intrastrand transfer reaction, the Figure-eight intermediate, is a substrate for integration. To test this, we introduced a nick into the junction at L-TIR on the non-transferred strand (R35(j6)L35nick-5′OH; Fig 3A). Both TIRs were integrated into supercoiled DNA (lanes 10–12) but with reduced overall efficiency relative to R35(j6)L35nick-3′OH (lanes 16–18). Furthermore, the kinetics of the reaction were altered. While we could detect a product at an early time point with R35(j6)L35nick-5′OH, subsequent accumulation of product was much slower. It is possible that the reaction is compromised due to the double-stranded break upon cleavage at the L-TIR. A 1 nt recession of the non-transferred strand at L-TIR (R35(j6)L35-5′rec) decreased integration activity to a barely detectable level (Fig 3A, lanes 19–21).

To probe the role of the target substrate, we repeated the junction integration experiments with linearized rather than supercoiled pUC19 as the target (Fig 3B). Although we expected a smear of

   

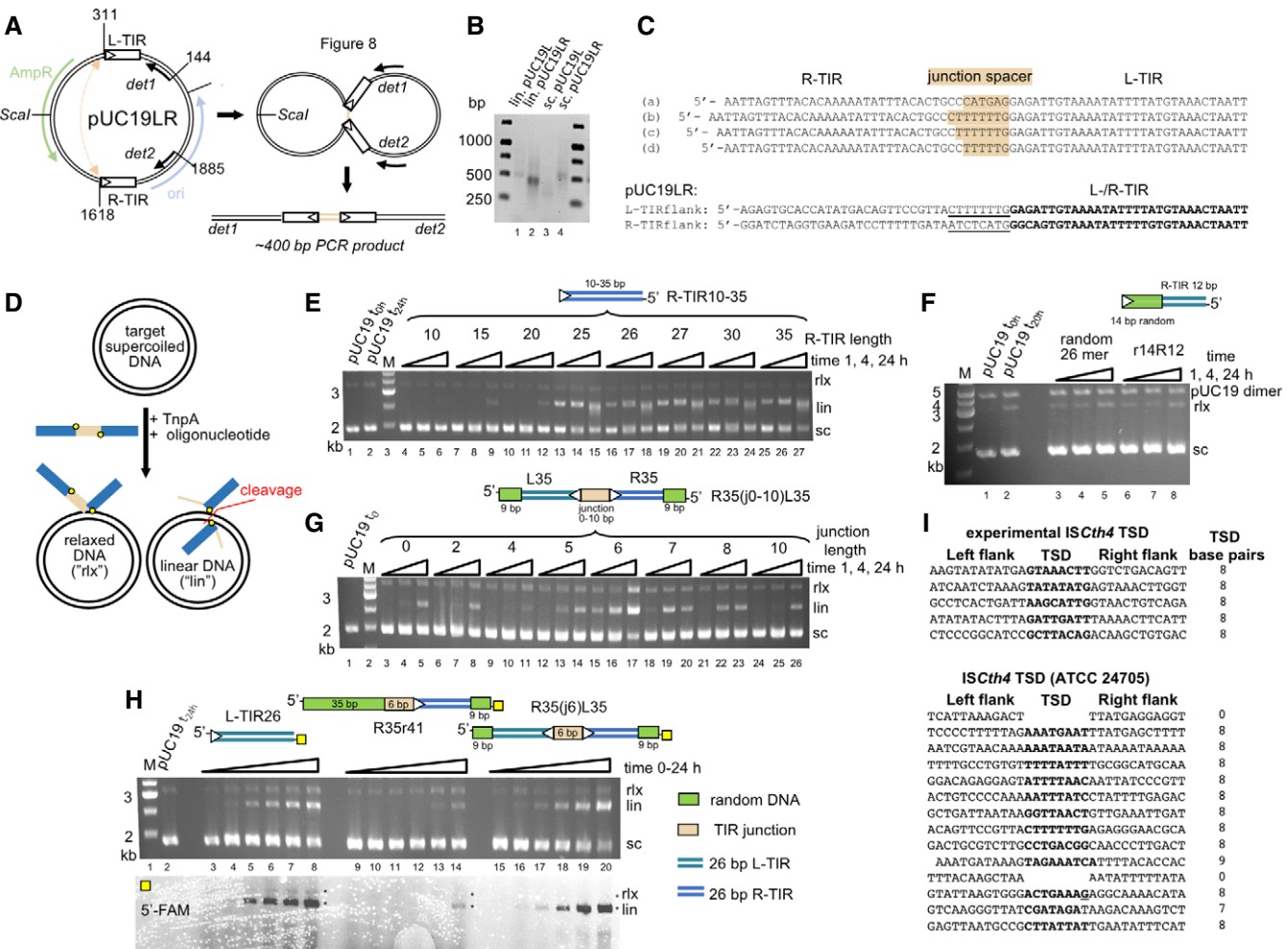

**Figure 2. Characterization of IS*Cth4* transposase strand transfer reactions.**

A   Schematic of assay to detect formation of Figure-eight intermediate. pUC19 was modified to include the 35-bp L-TIR and R-TIR sequences (pUC19LR) or L-TIR only as a control (pUC19L). White triangles mark the TIR ends. Strand transfer of one TIR to the other (depicted as orange arrow) was detected using PCR primers *det1* and *det2* (Appendix Table S1).

B   Detection of strand transfer by PCR. pUC19L or pUC19LR was used as substrate in supercoiled (sc.) or linear (lin.) form. Activity as assessed by a PCR product of expected size was detected when pUC19LR was linearized (lane 2). Other reactions yielded only background products of incorrect sizes.

C   (top) Sequences of four unique Figure-eight junction intermediates, obtained by cloning the PCR band in lane 2 of panel B, and screening 24 colonies, eight of which corresponded to junctions. Presented sequences are the reverse complements of those detected. (bottom) For reference, the 8-bp sequences adjacent to each cloned TIR in pUC19LR are underlined.

D   Schematic of assay to detect single-end (rlx) and double-end (lin) joined products. The target plasmid is supercoiled pUC19 (sc), and the reaction shown depicts junction integration. TIR sequence is in blue, junction spacer sequence in orange. Yellow dots indicate the 3′-OH groups.

E   *In vitro* integration of R-TIR as a function of TIR length. Lane 1, pUC19 alone at $t = 0$. Lane 2, pUC19 after incubation for 24 h in reaction buffer omitting only the TIR oligonucleotide. White triangle marks the TIR end.

F   Comparison of *in vitro* integration of a random 26-mer (lanes 3–5) and an oligonucleotide where 14 bp of the R-TIR were replaced by random sequence (in green); bp 13–26 of the TIR are unchanged (lanes 6–8). For both substrates, the low level of relaxed plasmid formation after 24 h is similar to that of pUC19 alone after 20 h of incubation in reaction buffer when only the TIR oligonucleotide is omitted (lane 2). White triangle marks the TIR end.

G   *In vitro* integration of transposon junction mimic as a function of junction length (depicted in orange). Green indicates 9 bp of random DNA added to oligonucleotide ends to direct correct annealing. The color scheme is maintained throughout. Lane 1, pUC19 alone at $t = 0$. White triangles mark the TIR ends.

H   Comparison of *in vitro* integration of L-TIR26 (lanes 3–8), R-TIR flanked by 41 bp of random DNA (R35r41, lanes 9–14), and TIR junction mimic with 6-bp T/A-rich spacer (R35(j6)L35, lanes 15–20). For each reaction shown, time points are as follows: 0, 0.5, 1, 2, 5, and 24 h. Lane 2, pUC19 after incubation for 24 h in reaction buffer omitting only the TIR oligonucleotide. (top) Ethidium bromide-stained gel. (bottom) Same gel visualized by fluorescence detection. The yellow square indicates the location of 5′-FAM. White triangles mark the TIR ends.

I   (top) Identified target sites in pUC19 with the central 8 bp in bold corresponding to the sequenced linear products from the reaction in panel G (lane 17). (bottom) Target sites from *Clostridium thermocellum* (ATCC 24705) show a similar preference for A/T-rich sequences.

Data information: "M", base pair marker with indicated size of standards. Labels rlx, lin, and sc mark positions of relaxed, linear, and supercoiled plasmid forms, respectively.

Source data are available online for this figure.

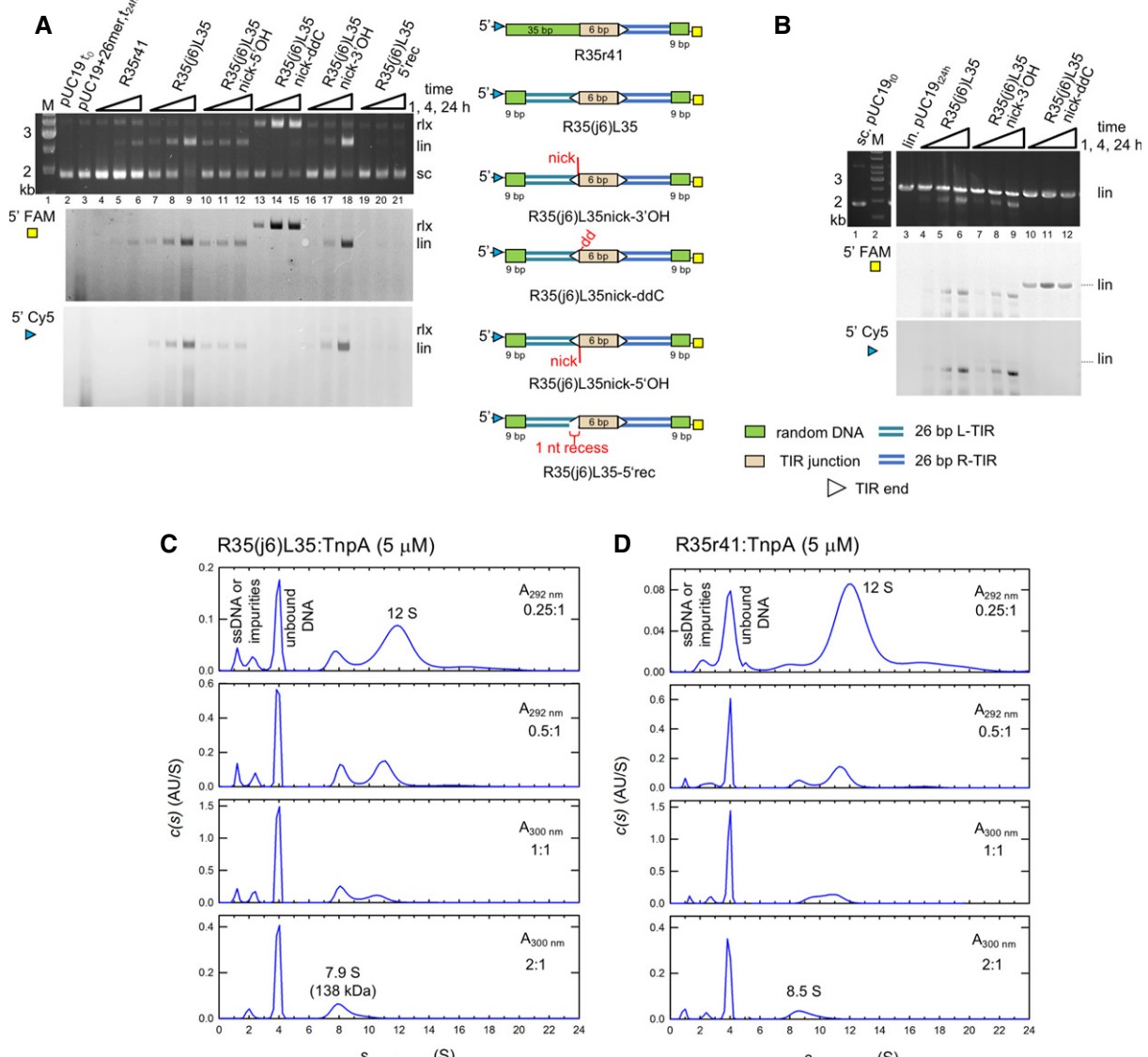

**Figure 3. Characterization of TIR junction integration and sedimentation analysis of DNA:TnpA complexes.**

A *In vitro* integration of TIR junction mimics. Modifications to R35(j6)L35 are as shown schematically on the right. The 5′-end of the L-TIR was labeled with Cy5 (blue triangle) and the 5′-end of the R-TIR with FAM (yellow square). The DNA color code corresponds to that in Fig 2. Lane 2, pUC19 alone at *t* = 0. Lane 3, pUC19 after incubation for 24 h with non-related 26-mer with one 5′-OH FAM label. White triangles mark the TIR ends.

B *In vitro* integration of oligonucleotide junctions using a linearized pUC19 target. Lane 1, supercoiled pUC19 alone at *t* = 0. Lane 3, pUC19 after incubation for 24 h in reaction buffer omitting only the TIR oligonucleotide.

C Sedimentation velocity analytical ultracentrifugation (SV AUC) analysis of the complex formed between TnpA and junction oligonucleotide R35(j6)L35 as a function of increasing DNA concentration.

D SV AUC analysis of the complex formed between TnpA and the 35-bp R-TIR flanked by 41 bp of random sequence, R35r41, as a function of increasing DNA concentration. The difference in *s*-values between the two complexes (7.9 vs. 8.5 S) may reflect differences in the stability of the complexes with different DNAs in the timescale of the experiment.

Data information: "M", base pair marker with indicated size of standards. Labels rlx, lin, and sc mark positions of relaxed, linear, and supercoiled plasmid forms, respectively.

Source data are available online for this figure.

products of varying size due to non-specific double-ended integration as seen in Fig 3A, instead we observed the accumulation of a fluorescently labeled ∼ 2.4-kb product with both R35(j6)L35 and R35(j6)L35nick-3′OH (lanes 4–9), suggesting a preferred insertion site. When compared to the supercoiled target where almost 100% conversion to a linearized product was achieved in 24 h (Fig 3A, lane 9), the reaction with a pre-linearized target substrate was much slower since only a portion of linear target was converted (Fig 3B,

lane 6). In contrast, the reaction with R35(j6)L35nick-ddC (lanes 10–12) did not result in double-ended cleavage; the labeled linear product migrates at the same position as the linear substrate indicating that only a single end was integrated, as seen with supercoiled target. Thus, a supercoiled target stimulates TIR junction integration. Such an effect of DNA supercoiling on transposition has been described before, for example, in the case of the MuA system where it affects many aspects of the transposition reaction (Naigamwalla & Chaconas, 1997; Manna & Higgins, 1999) and Tn5 which also has integration specificity for supercoiled target DNA (Lodge & Berg, 1990). The eukaryotic Hsmar1 transposase of the Tc1/mariner family has similarly been shown to prefer supercoiled targets (Bouuaert & Chalmers, 2013).

## TIR ligands and TnpA form complexes with 1:2 stoichiometry when excess DNA is present

Under physiological salt concentrations (i.e., ~ 150 mM), the highest achievable concentration of TnpA alone was ~ 3 μM (~ 0.15 mg/ml), above which the protein aggregated. Sedimentation velocity analytical ultracentrifugation (SV AUC) at 0.5–2 μM (Appendix Fig S3A) showed the presence of two species with sedimentation coefficient values of 3.4 S and 4.8 S. The concentration dependence of the relative ratio of the two is typical of reversible self-association. To determine the molecular weight of the smallest species, sedimentation equilibrium analytical ultracentrifugation (SE AUC) was conducted at a loading concentration of 0.25 μM TnpA. The results are consistent with a single species of $M_w = 41 \pm 2$ kDa (Appendix Fig S3B and C), in good agreement with the calculated TnpA monomer mass of 47 kDa. This most likely corresponds to the 3.4 S species observed in SV AUC. Thus, at < 0.5 μM, TnpA is predominantly monomeric and with increasing concentration the transposase dimerizes as evidenced by the 4.8 S species.

When TnpA was bound to TIR DNA oligonucleotides, we observed an improvement in solubility and a concentration dependence in the oligomerization state of complexes. As described in more detail below, in the concentration range used for SV AUC (0.5 or 5 μM) and with molar excess of DNA, TnpA forms complexes in which a dimer is bound to one DNA molecule for all the TIR substrates we tested. However, in the concentration range of 20–40 μM (~ 1–2 mg/ml) that we used to analyze sample monodispersity with size-exclusion chromatography (SEC), the results were more complicated as we encountered a range of multimeric states that were dependent not only on protein concentration but also on DNA:protein ratio.

In the case of blunt-ended TIR substrates at 0.5 μM TnpA and a 4:1 DNA:TnpA ratio, the complex with R-TIR26 (7.2 S by SV AUC) yields a best-fit $M_w$ of 115 kDa (Appendix Fig S4A), consistent with a TnpA dimer bound to one DNA molecule (theoretical $M_w = 110$ kDa), and excess unbound DNA remained. In contrast, when more concentrated TnpA was mixed with R-TIR26 (Appendix Fig S3D; 20 μM TnpA) or L-TIR26 (Appendix Fig S3E; 40 μM TnpA) at different DNA:protein ratios and analyzed by SEC, a discrete high $M_w$ ~ 450 kDa complex formed that corresponds to an octamer bound to four TIRs. We observed no difference in behavior when we used R- or L-TIR substrates, suggesting that they have similar affinities and consistent with our observation that either end can serve as the donor TIR.

Similar results were obtained when we evaluated oligonucleotides mimicking substrates along the transposition pathway such as the TIR junction (R35(j6)L35; Fig 3C and Appendix Fig S3F) or the R-TIR flanked by random DNA (R35r41; Fig 3D and Appendix Fig S3G). Using SV AUC, 5 μM TnpA mixed with either R35(j6)L35 (Fig 3C) or R35r41 (Fig 3D) in a DNA:TnpA ratio of 0.25:1 (i.e., excess protein) yielded mixture of complexes at ~ 7.9 S and ~ 12 S in addition to unbound DNA. The smaller R35(j6)L35:TnpA complex at 7.9 S (best-fit $M_w$ = 138 kDa) corresponds to a dimer of TnpA bound to one oligonucleotide which has a predicted $M_w$ for the complex of 141 kDa. The reaction boundary at ~ 12 S presumably reflects a tetramer although with unclear DNA:protein stoichiometry. As the relative ratio of DNA was increased, the ~ 12 S boundary disappeared and only that at ~ 7.9 S was observed. In contrast, at a higher TnpA concentration of 40 μM, we observed complexes with heterogeneous elution profiles using SEC. The peaks corresponding to multiple oligomeric states were dependent on the DNA:protein ratios, and the same overall profile was observed if the spacer was followed by the second TIR (Appendix Fig S3F) or by random DNA (Appendix Fig S3G).

In these SEC and SV AUC experiments, there was no evidence for monomeric TnpA binding to DNA, suggesting either that TnpA only binds DNA as a dimer or that it rapidly dimerizes when one monomer binds DNA. Since our biochemical integration assays were carried out in excess of DNA (2 μM) relative to TnpA (0.25 μM), a condition where we only detect monomeric TnpA or a 1:2 DNA:protein complex, we conclude that a 1:2 complex is the active state of TnpA regardless of the type of DNA substrate. Although we cannot completely rule out alternate explanations, it seems likely that the large multimers present in excess of TnpA or at higher (> 20 μM) concentrations are artifacts due to aggregation properties of TnpA under the specific conditions used.

## TnpA forms asymmetric complexes with DNA

Using X-ray crystallography, we determined the structures of three different complexes of TnpA bound to DNA, reflecting different steps along the copy-out/paste-in transposition pathway (Table 1). The pre-reaction complex (PRC) is a complex of TnpA with DNA representing the first 26 bp of the R-TIR with six flanking bp (fR-TIR26; Fig 4A); the pre-cleaved complex (PCC) is a complex with the first 26 bp of the R-TIR (R-TIR26; Fig 4B and D); and strand transfer complex 1 (STC1) is a complex with oligonucleotide R26 (j6)L15-5′rec in which a 6-bp bridging spacer links two TIRs (Fig 4C) and is an approximation of the Figure-eight intermediate. Representative composite-simulated annealed omit maps are presented in Appendix Fig S7A–C.

In all three complexes, a single DNA molecule is bound by a dimer of TnpA and they are all remarkably asymmetric (Fig 4). In each structure, the R-TIR is recognized through an extensive protein–DNA interface (Fig 5) in which almost all of the interactions are contributed by one protomer ("A"; Fig 4D) with the exception of the transposon tip which is directed toward the catalytic domain ("CD"; residues 165–261 + 325–407) of the second protomer ("B"), suggesting that catalysis occurs *in trans*. In the PCC (Fig 4B), the 3′-OH of the transferred strand (TS) points toward the active site, a position consistent with donor TIR binding rather than recipient TIR binding. Using the PCC, we first describe the general features of the

**Table 1. Crystallographic statistics.**

| | Se-met PCC A | Se-met PCC B | Se-met PCC C | PCC | PRC | STC1 |
|---|---|---|---|---|---|---|
| Data collection | | | | | | |
| X-ray source | APS ID-22 | APS ID-22 | APS ID-22 | APS ID-22 | APS ID-22 | CuKα |
| Detector | MARCCD | MARCCD | MARCCD | MARCCD | EIGER 16M | Saturn A200 |
| No. of crystals merged | 1 | 1 | 1 | 1 | 1 | 2 |
| Space group | P2$_1$2$_1$2$_1$ | P2$_1$2$_1$2$_1$ | P2$_1$2$_1$2$_1$ | P2$_1$2$_1$2$_1$ | P2$_1$2$_1$2$_1$ | P3$_1$2 |
| Cell dimensions | | | | | | |
| a, b, c (Å) | 88.7, 100.7, 154.3 | 88.8, 100.7, 154.5 | 88.8, 100.7, 153.9 | 89.68, 109.50, 157.70 | 89.58, 99.05, 156.09 | 114.6, 114.6, 232.6 |
| α, β, γ (°) | 90.0, 90.0, 90.0 | 90.0, 90.0, 90.0 | 90.0, 90.0, 90.0 | 90.0, 90.0, 90.0 | 90.0, 90.0, 90.0 | 90.0, 90.0, 120.0 |
| Wavelength (Å) | 0.979493 | 0.979243 | 0.933265 | 1.000 | 1.000 | 1.5418 |
| Resolution (Å) | 3.9 | 4 | 4 | 3.5 | 3.5 | 3.5 |
| $R_{merge}$ (%)[a] | 7.4 | 8.1 | 5.8 | 7.0 | 8.0 | 17.0 |
| $I/\sigma(I)$ | 11.2 (0.6) | 10.4 (0.8) | 14.5 (1.8) | 12.2 (1.99) | 9.7 (2.06) | 14.5 (1.93) |
| Number of measurements | 94,530 | 87,501 | 86,701 | 144,992 | 65,615 | 494,599 |
| Unique data | 13,092[c] | 12,156[c] | 12,098[c] | 20,123[d] | 18,004[d] | 22,945[d] |
| Completeness (%) | 99.3 | 99.85 | 99.78 | 99.8 (100) | 99.4 (99.7) | 99.8 (100) |
| Redundancy | 7.2 | 7.2 | 7.2 | 7.2 | 3.6 | 21.5 |
| Maximum likelihood phasing | | | | | | |
| Anomalous resolution (Å) | 6.6 | 6.1 | 7.2 | – | – | – |
| No. of sites | 18 | 18 | 18 | – | – | – |
| Anomalous completeness (%) | 99.8 | 99.9 | 99.8 | – | – | – |
| Refinement | | | | | | |
| Resolution (Å) | – | – | – | 3.5 | 3.5 | 3.5 |
| No. reflections | – | – | – | 20,102 | 17,950 | 22,938 |
| $R_{work}/R_{free}$ (%)[b] | – | – | – | 22.1/26.7 | 24.9/29.6.0 | 26.1/29.7 |
| No. atoms Protein/DNA | – | – | – | 6,127/1,060 | 6,159/1,289 | 6,166/1,896 |
| Average B-factor (Å$^2$) | – | – | – | 167.0 | 154.0 | 160.0 |
| R. m. s. d. bond lengths (Å) | – | – | – | 0.004 | 0.003 | 0.003 |
| R. m. s. d. bond angles (°) | – | – | – | 0.673 | 0.571 | 0.572 |
| Ramachandran plot (% disallowed, allowed, favored) | – | – | – | 0.13/2.69/97.18 | 0.13/4.40/95.47 | 0.0/2.80/97.20 |
| PDB code | – | – | – | 6XG8 | 6XGW | 6XGX |

R. m. s. d., root mean square deviation.

[a] $R_{merge} = \Sigma|I_i - \langle I \rangle|/\Sigma I_i$, where $I_i$ is the intensity of measured reflection and $\langle I \rangle$ is the mean intensity of all symmetry-related reflections.

[b] $R_{free} = \Sigma T||F_{calc}| - |F_{obs}||/\Sigma F_{obs}$, where $T$ is a test dataset of about 3–6% of the total unique reflections randomly chosen and set aside prior to refinement.

[c] Friedel's law false.

[d] Friedel's law true.

assemblies, and later, we consider the mechanistic implications from the comparison of the three structures.

**DNA-bound TnpA forms an asymmetric dimer**

Three small N-terminal domains precede the catalytic domain of TnpA: a dimerization domain ("DD"; residues 1–56), an N-terminal DNA-binding domain ("NDB"; residues 56–108), and a helix-turn-helix domain (HTH; residues 108–165). These N-terminal domains are responsible for dimerization and subterminal DNA binding, all playing a part in generating the asymmetry of the DNA-bound TnpA dimer.

The TnpA dimer is held together largely by two dimerization interfaces. The most extensive buries an overall interface of

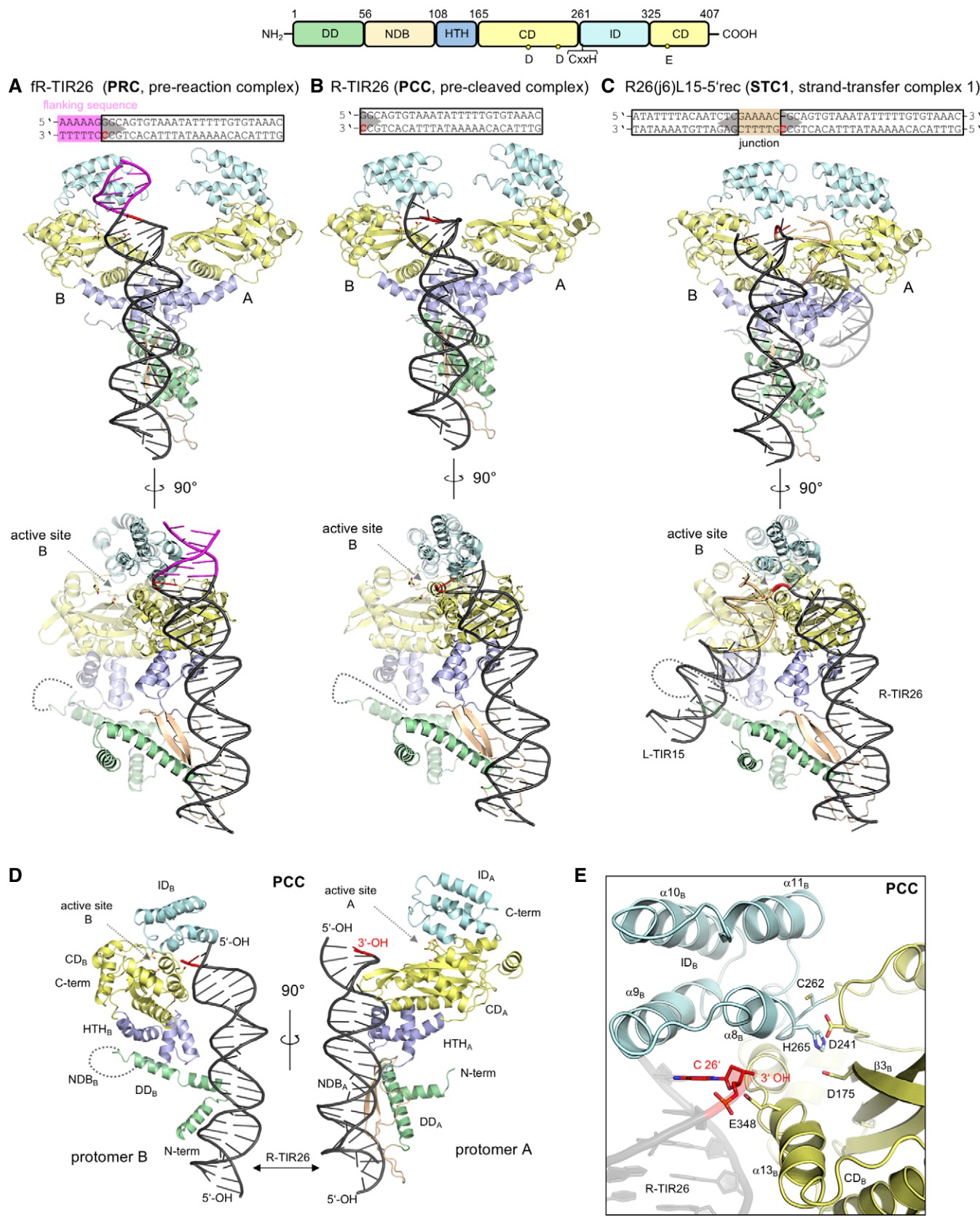

**Figure 4.**

**Figure 4.  Structures of ISCth4 TnpA bound to oligonucleotide substrates.**

A–C    Structures of the complexes between TnpA and R-TIR with 6 bp of flanking DNA (PRC); R-TIR26 (PCC); and a junction mimic consisting of 26 bp of the R-TIR, 15 bp
of the L-TIR, and a 6-bp junction (STC1). For clarity, protomer B is transparent in the 90° side views (bottom). The domain color scheme is indicated (top). Active
site residues in protomer B are shown as sticks (D175, D241, E348).

D    Domain organization of TnpA protomers and their mutual position relative to R-TIR26 in PCC. DD, dimerization domain. NDB, N-terminal DNA-binding domain.
HTH, helix-turn-helix domain. CD, catalytic domain. ID, insertion domain. NDB$_B$ is disordered and represented with a dashed curve. The location of the active site is
marked with an arrow. Active site residues in protomer B are shown as sticks (D175, D241, E348).

E    Close-up view of the active site in the PCC with CD$_B$ in yellow and the ID in cyan. The positions of the DDE residues (D175, D241, and E348) converging to the
terminal nucleophilic 3′-OH of R-TIR26 and the CxxH motif (C262 and H265) are shown as sticks. The terminal bp, C26, is highlighted in red.

~ 1,400 Å$^2$ (calculated using PISA, Krissinel & Henrick, 2007) and is formed by the mutual interactions between the two DD domains that consist of the first two N-terminal α-helices of each protomer (Fig 5D and E). Although α1 and α2 are leucine-rich, they lack the characteristic heptads of a leucine zipper motif (Appendix Fig S2) and do not form coiled-coil interactions typical of a leucine zipper such as that identified as the dimerization element in Tnp from IS911 (Haren et al, 2000). The two DDs are not structurally identical: α2$_A$ (subscript indicates which protomer) is almost straight whereas α2$_B$ is bent by 30° (Fig 5E). A smaller dimerization interface (~ 700 Å$^2$, Krissinel & Henrick, 2007) is formed by helices α3 and α4 of the HTH domains (Fig 5D).

The most startling difference between the protomers is that the NDB is completely disordered in one protomer (Fig 4D, protomer B) and not detectable in the electron density. As SDS–PAGE analysis of dissolved PCC crystals indicated the presence of only full-length TnpA (Appendix Fig S1D), the lack of density for NDB$_B$ was not due to proteolytic degradation but rather to the spatial constraint resulting from an unusual asymmetric packing of the N-terminal domains at the dimer interface. Specifically, HTH$_B$ packs directly against kinked helix α2$_B$ of DD$_B$, whereas the HTH domain of the protomer that is bound to DNA (HTH$_A$) packs against β–loop$_A$ of the visible NDB$_A$; NDB$_A$ in turn packs against the straight helix α2$_A$ of DD$_A$ (Fig 5D). The consequence of this differential domain packing is the lack of space and the necessary protein–protein interactions for NDB$_B$ to fold. The NDBs span residues 56–108, and the different distances between bordering amino acids in the two protomers, 19 Å for protomer B and 30 Å for protomer A with the folded NDB, reflect the scale of the dimer asymmetry. The asymmetry in the dimer interface packing is particularly striking when one considers the environments of hydrophobic side chains I136, Y137, and F139 of the HTH domains, which are tightly buried yet in very different packing environments in the two protomers (Fig 5D). Given these structural features, it is very difficult to imagine how the dimer could "symmetrize" itself to allow the binding of two TIRs in the same way.

HTH domains are predicted to be present in all copy-out/paste-in transposases that have been studied to date (Stalder et al, 1990; Nagy et al, 2004), and the structures here indicate that they play a crucial role in DNA recognition. The HTH domain of TnpA is a tri-helical motif (α3, α4, and α5; Figs 4D and 5D), and all three α-helices contribute residues involved in binding the terminal region of the R-TIR (Y120, S125, T126, R127, S147, E144, and K148; Fig 5A and B). A further consequence of the asymmetric dimer interface packing is that HTH$_A$ and HTH$_B$ are not related to each other by twofold rotational symmetry yet they still pack against each other. The regions of HTH$_A$ that bind the TIR are also available

in HTH$_B$ as they are on the dimer surface; however, as we have never detected complexes with two TIRs bound by a TnpA dimer, these are clearly not sufficient on their own to bind another TIR under the experimental conditions we have tested. It seems likely that the lack of a folded NDB$_B$ is why a second TIR is not bound to the dimer. Considering the asymmetric packing of the two HTH domains, any DNA binding by HTH$_B$ might be different than that observed for HTH$_A$ (see Discussion).

**The TnpA catalytic domain is a RNase H-like domain with a helical insertion**

The catalytic domain (CD) of TnpA has the RNase H-like fold with a DDE-type active site, and the active site residues D175, D241, and E348 are in a shallow cavity (Fig 4E). Although the resolution of the structures is limited and none of them contains metal ions in the active sites, in the PCC and PRC, the side chains of D175 and D241 are in the appropriate conformation for metal binding and catalysis (Appendix Fig S7D and E) whereas the side chain of E348 is pointing away from the presumed metal-binding site. In STC1, the active site is further disordered and the D175 side chain is turned in the opposite direction to that observed in the PCC and PRC structures (Appendix Fig S7F).

The CD is interrupted between strand β7 and helix α12 by an α-helical insertion domain ("ID"; residues 261–325) consisting of four α-helices, α8–α11 (Fig 4E). All-α-helical insertion domains have been observed in certain RNase H-like eukaryotic DNA transposases including in the hAT transposase Hermes (Hickman et al, 2014), Transib (Liu et al, 2019), the P element (Ghanim et al, 2019), and the related RAG1 recombinase (Kim et al, 2015). To the best of our knowledge, ISCth4 TnpA is the first structurally characterized prokaryotic transposase with an α-helical ID domain (Appendix Fig S5A). In the case of Hermes, X-ray structures showed that the histidine of the C/DxxH motif directly contacts the scissile phosphate and is an integral part of the active site (Hickman et al, 2018). Three-dimensional alignment of the CD/IDs of ISCth4 and Hermes indicates that the CxxH motifs are in the same position (Appendix Fig S5B; Hickman et al, 2014). Conserved histidines forming H-bonds with scissile phosphates have also been seen in strand transfer complexes of RAG1 and Transib (Liu et al, 2019; Chen et al, 2020).

In the structures here, CD$_A$ and CD$_B$ are almost identical and can be superposed with a ~ 1 Å r.m.s.d. However, due to the overall asymmetry of the complexes resulting from the arrangement of N-terminal domains, the two CD/IDs are not spatially related to each other by a pure rotation but are tilted relative to each other and the axis of the R-TIR (Appendix Fig S4B and C).

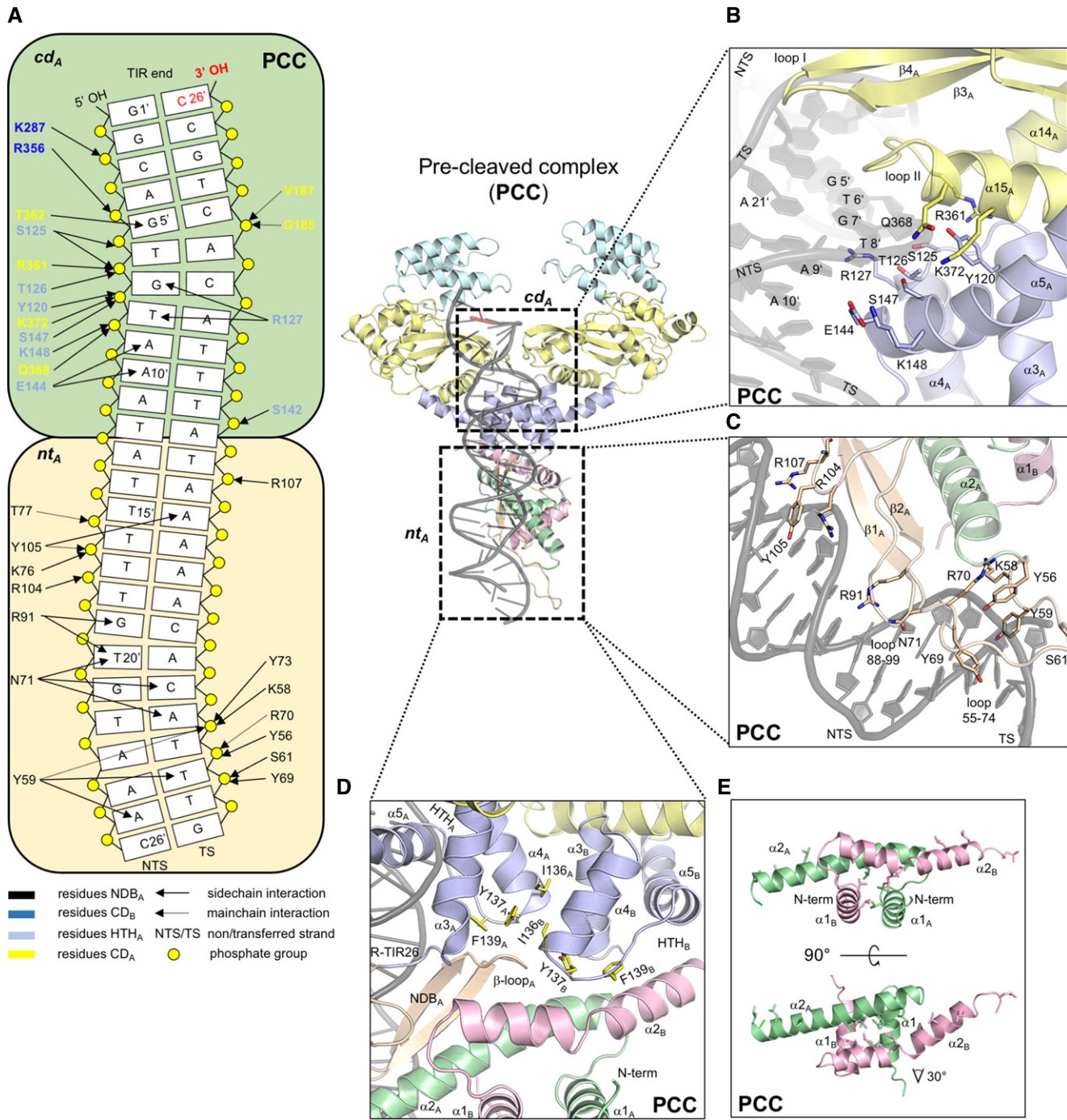

**Figure 5. Features of the complex between IS*Cth4* TnpA and its pre-cleaved R-TIR (PCC).**

A　Diagram of the observed interactions between TnpA and R-TIR26 in PCC. $cd_A$ is the binding site on TnpA assembled from $CD_B$, $HTH_A$, and elements of $CD_B$. $nt_A$ is the binding site on TnpA consisting of residues from $NDB_A$. Blue arrows depict interactions between R-TIR26 and protomer B.

B　Close-up view of terminal R-TIR26 region (bp 5–10) bound by $cd_A$ in the PCC.

C　Close-up view of interactions in the $nt_A$-binding site in the PCC.

D　Close-up view of the packing at the N-terminal part of TnpA dimer in the PCC.

E　Close-up view of leucine-rich dimerization domains in the PCC.

Data information: Color scheme is maintained from Fig 4. For clarity, dimerization domain of protomer B is depicted in pink.

## Bipartite binding to the donor TIR

In all three complexes, R-TIR26 is bound by two binding sites, designated $nt_A$ and $cd_A$ (Fig 5A). In the $nt_A$-binding site, $NDB_A$ forms a rich set of interactions with the DNA between subterminal base pairs 14–25 through adjacent minor–major–minor grooves (Fig 5A and C). Y59 and Y105, approximately defining the ends of $NDB_A$, lie deep in the two minor grooves assisted by nearby K58, R70, R107, and R104 (Fig 5C). The two β-strands of the $NDB_A$ are in the major groove and form a number of base-specific interactions. The combined effect of a widened major groove and the narrowed adjacent minor groove due to Y105 is an approximately 30° bend of the DNA (Fig 4D). Intriguingly, based on the analysis of the DALI server, $NDB_A$ has no known structural homologs (Holm, 2019). Closer to the TIR tip, the $cd_A$-binding site is formed by residues from both the HTH and CD domains of protomer A (Fig 5A) as well as by elements of the CD domain of protomer B ($CD_B$) and contacts terminal base pairs 1–12. All three helices of the $HTH_A$ domain contact the DNA with the N-terminal ends of α4 and α5 in the major groove where they form extensive non-specific and base-specific interactions (Fig 5A and B). The importance of some of these protein–DNA interactions has been demonstrated for the IS256 transposase where mutation of any of the conserved residues Y111 (Y120 in TnpA), G114 (G123 in TnpA), T117 (T126 in TnpA), or R118 (R127 in TnpA) abolished or significantly reduced DNA binding (Hennig & Ziebuhr, 2010).

One of the most important roles of the $cd_A$-binding site is to direct the tip of the transposon into the active site of $CD_B$, and presumably to position it appropriately for catalysis (Figs 4D, and 5A and B). This is accomplished by two structural elements of $CD_A$, the β-hairpin between its two-first β-strands (β3 and β4; "loop I") and a second loop (residues 360–368, "loop II"), which significantly widen the minor groove close to the transposon tip (Fig 5B), causing the DNA to bend about 45°. The regions of the TIR that are contacted by TnpA are largely conserved between the L- and R-TIRs of IS*Cth4* (Fig 1D). The same DNA-binding elements described here for $CD_A$ are available in $CD_B$ yet there is no DNA bound to $CD_B$ (Appendix Fig S4B and C), presumably due to the absence of a folded $NDB_B$.

## Comparison of the three complexes suggests how Figure-eight intermediates are formed

All three complexes display the same asymmetric dimerization, suggesting it is an inherent property of TnpA when bound to DNA. While the quality of the electron density maps differs, the organization of the DD and NDB domains is essentially identical in the three complexes, and in all three structures, bp 5–26 of the R-TIR are recognized in the same way. Thus, the presence or absence of the flanking sequence does not have an appreciable effect on donor TIR binding. On the other hand, there are significant differences in the relative positions of the CD and ID domains and the tips of the TIR (Fig 4A–C; Appendix Fig S7D–F).

In the PRC, representing the state in which the flanking sequence is still present and before any chemical step has occurred, the transposon tip is away from the active site of $CD_B$ as helix α9 of the ID is inserted in the major groove at the flanking region (1–5 base pairs from the TIR end), where it interacts with the DNA through

K280, R284, K287, and R288. The scissile phosphate in fR-TIR26 is displaced ~ 8 Å from the position of the R-TIR26 3′-OH in the PCC structure (Fig 6A and B) and ~ 16 Å from the equivalent position of the scissile phosphate in Hermes when bound to DNA (Appendix Fig S7E, Hickman *et al*, 2018). When we superimposed the A protomers of the PCC and PRC structures (which can be done with an overall r.m.s.d. of ~ 1.1 Å over 400 Cα positions), the clear difference is a ~ 6 Å change in the position of $ID_B$ as it closes down on the transposon end in the absence of the flanking sequence (Fig 6B). The inherent mobility of the ID is supported by a normal mode analysis using iMODS (López-Blanco *et al*, 2014) in the absence of DNA (Fig 6C) which indicated that the ID can move relative to the CD. Furthermore, the primary mode of ID motion is consistent with the motion seen in the comparison of the two complex structures (Fig 6B). The comparison of the active site of PCC to the active site of Hermes when bound to DNA (Hickman *et al*, 2018) reveals that although the 3′-OH of the TIR end is in proximity to the catalytic residues, the scissile phosphate that would be present on the TIR end is still displaced from the equivalent position of the scissile phosphate in the Hermes complex by ~ 5 Å (assuming ideal geometry). This is likely due to a combination of factors including the lack of bound metal ion to organize the active site, stabilization of the terminal nucleotide by base pairing to the complementary strand, and the lack of flanking DNA that would be part of the authentic pre-cleaved state.

In contrast with the PRC, in STC1 the scissile phosphate is ~ 6 Å displaced relative to that observed in the assembled Hermes active site (Appendix Fig S7F). Strikingly, whereas in PRC the flanking DNA is directed away from the active site by $ID_B$, in STC1, the 6-bp DNA junction bridge between the L- and R-TIRs and the entire L-TIR are instead directed in the opposite direction with an overall bend of about 120° (Figs 4C and 6D). On the recipient L-TIR side of the spacer, there are DNA:TnpA contacts up to the eighth base pair from the gap, and the transposon end of L-TIR is ~ 30 Å away from the active site of $CD_B$ (Fig 6D and E). $HTH_B$, $CD_B$, and elements of $CD_A$ all contribute to the binding of spacer DNA and L-TIR although the interactions are less well-defined than those to the donor R-TIR and the quality of the electron density is poorer on the recipient side of the junction. We were able to capture this state crystallographically only after introducing a nick and a one nucleotide recess on the non-transferred strand of R-TIR (R26(j6)L14-5′rec). This likely relieved the stress on the DNA caused by bending and helped to stabilize the complex.

Relative to the other two complexes, in STC1, the $CD_B$ domain has shifted ~ 10 Å closer to $CD_A$, and $ID_A$ and $ID_B$ have formed a new protein–protein interface of ~ 990 Å$^2$, effectively enclosing the spacer DNA. The movement is a rigid body shift, and $CD_B$ can still be superposed on $CD_A$ with an r.m.s.d. of 1.1 Å over 301 Cα positions. The new interface is assembled from residues in the loops between helices α8–α9 and helices α10–α11 (Fig 6D and E). The spacer is bound by the now-closed ID domains near the new $ID_A$/$ID_B$ interface where R279 and R268 of $ID_B$ form non-specific interactions (Fig 6E and F), consistent with the non-specific sequences that flank copies of IS*Cth4* in *C. thermocellum*. Most of the interactions with the recipient L-TIR are provided by helix α4 of $HTH_B$ and α13 of $CD_A$ that, in comparison with PRC and PCC, are brought closer to each other, thereby creating an additional DNA-binding site ($cd_B$; Fig 6D and F).

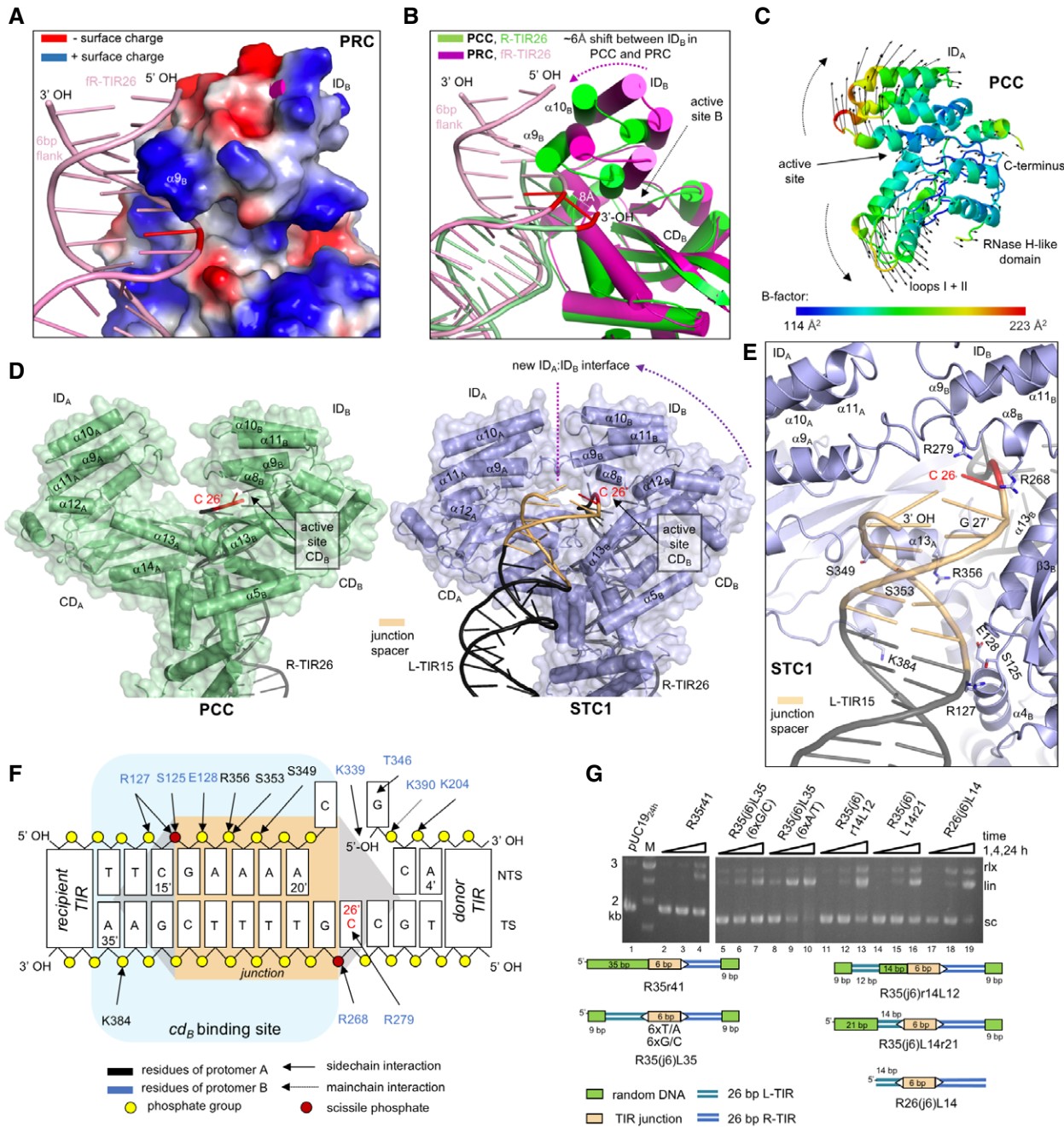

**Figure 6. Domain movements during Figure-eight formation.**

A  The position of helix α9$_B$ relative to the flanking DNA. TnpA is depicted in a surface charge representation.

B  ID$_B$ movement associated with binding flanking DNA. The double-headed white arrow shows the 8 Å shift between the positions of 3′-terminal residues (in red) of R-TIRs in the PRC and PCC.

C  Normal mode analysis of CD and ID domains of IS*Cth4* Tnp (residues 160–407) reveals the potential for relative motion (marked with black arrow field). Colors correspond to the crystallographic B-factor of the model.

D  Comparison of PCC and STC1 structures, backside view.

E  Close-up of *cd$_B$* in STC1 structure with bound junction spacer (in orange) and the tip of the recipient L-TIR (in gray). Labeled residue side chains shown as sticks belong to the *cd$_B$*-binding site.

F  Diagram of observed interactions between TnpA and tip of donor R-TIR, junction spacer and recipient L-TIR in STC1. Interactions with the rest of R-TIR are identical to those in the PCC.

G  *In vitro* integration of TIR junction mimics with modified L-TIRs or spacer sequence. Lane 1, pUC19 alone after 24 h of incubation in reaction buffer omitting only oligonucleotide substrate. White triangles mark the TIR ends. "M", base pair marker. Labels rlx, lin, and sc mark positions of relaxed, linear, and supercoiled plasmid forms, respectively.

Source data are available online for this figure.

Several features of STC1 suggest that it captures the state of the transposition reaction after the formation of the Figure-eight intermediate. The site of strand transfer is sequestered by the ID domains clamping down on it, perhaps to protect the junction (which now has nicks at or near both transposon ends) as it awaits the arrival of the replication machinery. The paucity of interactions with the L-TIR recipient is consistent with a product state, since the covalent link between the two TIRs has relieved the need to hold tightly onto both. Indeed, weak binding of the recipient end after junction formation may help TnpA relinquish its hold on the Figure-eight intermediate to allow the replication machinery access.

**Junction integration does not require two full-length TIRs**

As we observed very few interactions in STC1 involving the recipient TIR, we were curious if junction substrates with one short TIR could be integrated *in vitro*. As shown in Fig 6G, an asymmetric minimal TIR junction that lacks the 12 subterminal base pairs of the L-TIR (RE26(j6)L14, lanes 17–19) retained integration activity although with slower kinetics relative to the full-length TIR junction, R35(j6)L35 (lanes 5–10) indicating that junction integration is not dependent on two full-length TIRs. We also observed that bp 1–14 and 15–26 of the L-TIR independently contribute to the activity since when we compared two substituted TIR junctions in which specific L-TIR sequences were replaced with non-related DNA, both R35(j6)r14L12 (lanes 11–13) and R35(j6)L14r21 (lanes 14–16) retained similar activity.

To verify that both TIRs from a minimal junction were integrated, we carried out the integration reaction with both TIRs differentially labeled with FAM and Cy5 fluorescent tags (Appendix Fig S6). A minimal asymmetric junction (RE26(j6)L14, lanes 2–4) and an analogue containing a nick at the 3′-end of the R-TIR (RE26(j6)L14nick-3′OH, lanes 5–7) generated a linear product that had both TIRs integrated. The difference between Cy5 and FAM fluorescence band intensity suggests lower integration efficiency for L-TIR of RE26(j6)L14 (lanes 2–4). When the 3′-end terminal nucleotide at R-TIR was replaced by a dideoxynucleotide (RE26(j6)L14nick-ddC, lanes 8–10), as expected, integration was not observed. We also asked if a minimal substrate mimicking the Figure-eight intermediate (with either a nick or recess at the 5′-end of the R-TIR, R26(j6)L14nick-5′OH or R26(j6)L14-5′rec, lanes 11–16) could serve as a substrate. Interestingly, these reactions did not yield any integration products, in direct contrast to the full-length variant R35(j6)L35nick-5′OH (Fig 3A, lanes 10–12) where a portion of activity remained.

# Discussion

Among the various mechanisms employed by DDE transposases, the copy-out/paste-in pathway has been among the most elusive in revealing its structural foundation. The structures here provide key insights into two aspects of the ISCth4 transposase that directs the reactions that comprise copy-out/paste-in transposition. The first is that a string of N-terminal domains specifically recognizes the subterminal sequence of the TIR and directs the formation of a highly asymmetric DNA-bound dimer, the catalytically functional unit. The second is the characterization of a four-α-helix insertion domain whose coordinated movement with the RNase H-like catalytic domain likely controls the progression of the transposition reaction.

The structures indicate that, at the initial step of donor TIR recognition, DNA binding relies on two N-terminal domains, the $HTH_A$ domain that contacts base pairs 7–11 close to the transposon end and the extended $NDB_A$ domain that winds its way alongside base pairs 14–25 in the subterminal region (Figs 4D and 5C). The net result of this binding, combined with dimerization driven by the DD domains, is the formation of a highly asymmetric dimer bound to one TIR. To assemble this complex, a TnpA dimer could bind a single TIR or a TnpA protomer could first bind one TIR and then recruit a second protomer (Fig 7). We observed a monomer–dimer TnpA equilibrium in the absence of DNA, but the organization of the complexes is also consistent with a pathway in which the $HTH_A$ and $NDB_A$ from a single protomer first bind to the subterminal region of a single TIR and then dimerization occurs via the DD domains.

Aspects of transposon end recognition seen here may well extend to the much larger group of prokaryotic TEs that carry out copy-out/paste-in transposition. For example, DNA binding by ISCth4 TnpA is consistent with previous results obtained with the IS256 transposase that showed that protein residues 1–130 of its transposase (which would include the first two α-helices of its predicted HTH domain) were sufficient to bind transposon ends (Hennig & Ziebuhr, 2010). Other families of copy-out/paste-in transposases have different types of N-terminal domains yet always at least one predicted HTH domain (Prère *et al*, 1990; Stalder *et al*, 1990; Rousseau *et al*, 2010). It seems likely that the general organization and division of labor between transposase regions in copy-out/paste-in Tnps resemble those seen in our structures. For example, in the IS3 family, an N-terminal fragment, OrfAB[1–149], of the 382-residue IS911 transposase is sufficient to bind transposon ends as a dimer (Haren *et al*, 2000) and mutations in the HTH motif within this fragment disrupt binding (Rousseau *et al*, 2004). Footprinting analysis revealed that OrfAB[1–149] does not contact the first ~ 10 bp of either transposon end and protects only subterminal regions (Normand *et al*, 2001). Similarly, the N-terminal 17 kDa fragment of the IS30 transposase protects only bp 9–35 of its IS ends (Stalder *et al*, 1990).

To initiate transposition, a free 3′-OH group must be generated at the end of one of the TIRs. The PRC structure captures the binding state before the initial nicking step on the donor TIR, where we observed TnpA interacting with both the donor TIR and flanking DNA. Helix α9 of the ID is inserted into the major groove of the flanking DNA and directs the scissile phosphate away from the active site. The structures and the iMODS analysis suggest that the relative flexibility of the ID and CD domains may be key to a conformation change that transiently places the transposon end in the *trans*-active site of $CD_B$. If the TIR flanking sequence is bent underneath the ID (by analogy to the Hermes transposase; Hickman *et al*, 2018; Appendix Fig S5B), then the cleavage site could reach the active site as approximated in the PCC (Fig 7A and Movie EV1). The need for DNA bending flexibility may be the reason we were able to detect Figure-eight formation only when we used linearized plasmid as a substrate and not supercoiled (Fig 2B). It is also possible that, as seen for RAG1/2 of the V(D)J recombinase, DNA at the TIR ends must be melted or deformed to acquire the correct configuration in the active site (Ru *et al*, 2018; Chen *et al*, 2020).

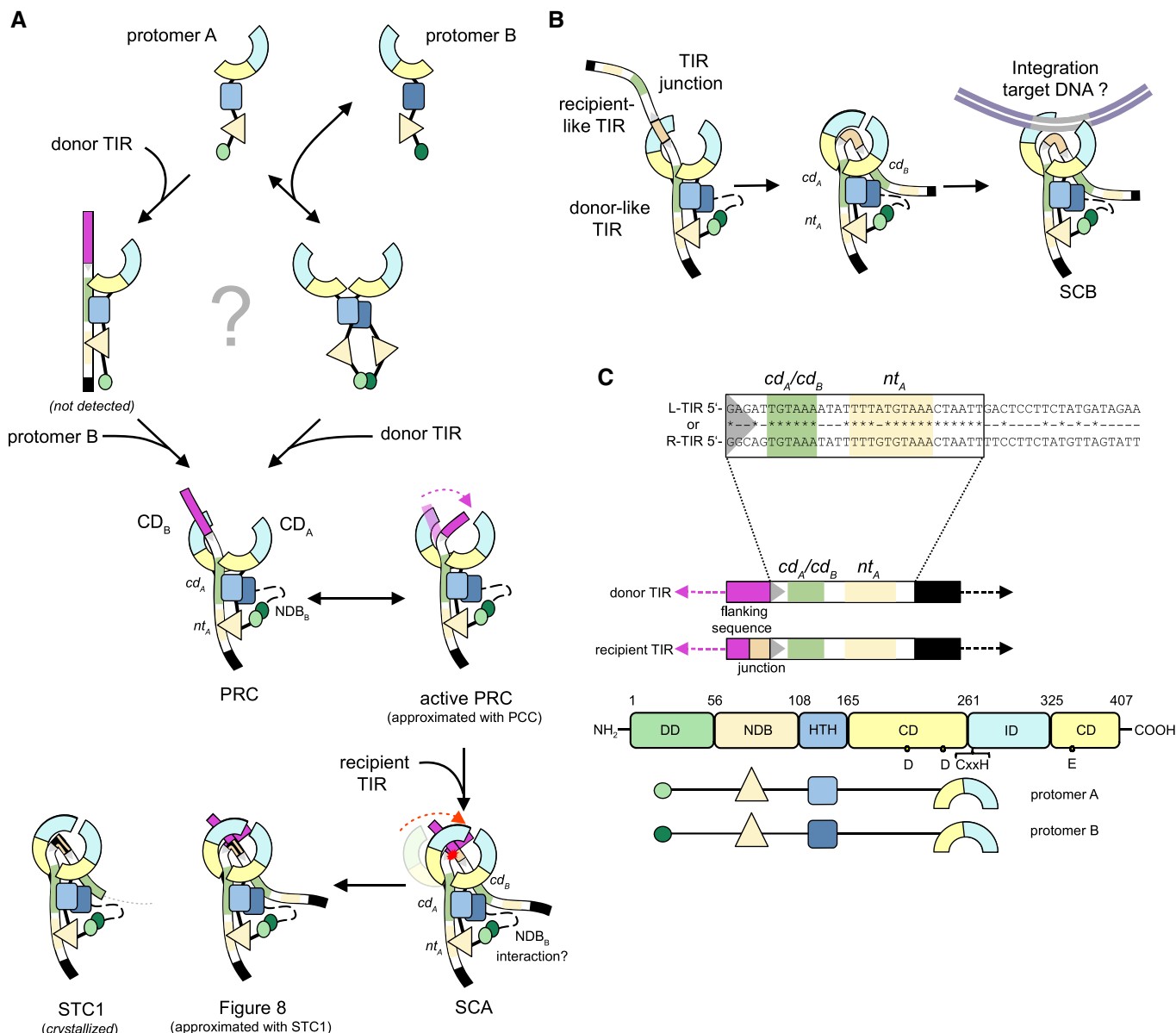

**Figure 7. Molecular model of copy-out/paste-in transposition.**

A Proposed model of Figure-eight formation. In solution, TnpA exists in monomer–dimer equilibrium. It is unclear if TnpA binds donor TIR as a monomer and then rapidly dimerizes or as a dimer. The resulting state is represented by the PRC structure which binds a TIR in an asymmetric bipartite manner with $nt_A$- and $cd_A$-binding sites and $NDB_B$ unstructured (indicated by the black dashed line). Given the range of motion of IDs, donor TIR end and flanking DNA may transiently bend under $ID_B$ (marked with the dashed magenta arrow) such that the TIR end is positioned into the active site. In this state (an active pre-reaction complex, approximated with the PCC structure, Movie EV1), the $CD_B$ could undergo a rigid body shift to close down on the TIR end (indicated by a red dashed arrow, Movie EV2) to generate the $cd_B$-binding site, bind the recipient TIR forming SCA, and catalyze the single strand transfer (marked with red star). How recipient TIR recognition is achieved is unclear but $NDB_B$ seems likely to participate. The resulting strand transfer complex contains the Figure-eight intermediate (the single-stranded junction is indicated by the discontinued orange line), a state approximated by the STC1 structure.

B Proposed model for TIR junction integration. TnpA binds to the TIR junction in the same way as in the PRC structure. By analogy to Figure-eight formation, the junction spacer may bend under $ID_B$, facilitated by either the $cd_B$-binding site and/or $NDB_B$ binding. This would position the donor-like TIR end into the active site of $CD_B$. Supercoiled target DNA (in gray) would then be captured for the next step of integration.

C Color schemes used in (A, B).

The model for Figure-eight intermediate formation (Fig 7A) suggests how an asymmetric dimer assembly might facilitate the strand transfer step of one transposon end into the flanking DNA of the second. The model builds on the third view captured of the

IS*Cth4* transposase in action, STC1, where we observed only limited and largely non-specific contacts between the recipient TIR and TnpA. After initial capture of the donor TIR, subsequent binding and recognition of the recipient TIR likely involves a rigid body shift

of CD$_B$ as documented in STC1 (Movie EV2) to generate an additional DNA-binding site for flanking DNA as well as the participation of either HTH$_B$ or NDB$_B$ of protomer B although the latter possibility would presumably require peeling apart the two monomers of the dimer to allow the second NDB room to fold. Due to the asymmetry imposed by binding to the donor TIR, flanking DNA rather than the transposon tip of the recipient TIR would be directed into the active site of CD$_B$. This provides an elegant explanation for strand transfer into flanking sequence with the concomitant generation of a spacer that is at the heart of circle junction formation.

Support for the model for Figure-eight intermediate formation is provided by studies of IS911 transposon circle formation as a function of deleting sequences at either transposon end (Normand et al, 2001). In these experiments, as the tip of one TIR was sequentially truncated, the length of the resulting spacer increased accordingly, suggesting that nucleophilic attack on the recipient end occurs at a fixed distance from the subterminal binding site and that intervening sequences are not important. A similar correlation was observed for IS2 when base pairs were deleted in a subterminal region (bp 13–19) of one end: the spacer length increased in step (Lewis et al, 2001). Several lines of evidence also suggest that the recipient TIR is bound less tightly and precisely than the donor TIR. First, Figure-eight junctions that we characterized (Fig 2C) and that have been reported for IS256 (Loessner et al, 2002) are of non-uniform length, suggesting a degree of imprecision. Figure-eight formation was also very inefficient in our hands and could only be detected by PCR, suggesting that interactions with recipient TIR, and therefore assembly of SCA, may be transient; this is consistent with footprinting results for IS2 that suggested the recipient TIR is bound only intermittently (Lewis et al, 2012). Finally, AFM studies on OrfAB[1-149] from IS911 have shown that the N-terminal domain alone is sufficient to assemble a synaptic complex containing two TIRs and that these are arranged in a parallel manner, precisely as proposed by the model in Fig 7A (Rousseau et al, 2010).

Despite the prevailing notion that integration of the dsDNA TIR junction is mediated by a symmetric complex SCB (Lewis et al, 2012; Chandler et al, 2015), it is possible that the structural asymmetry we observed in structures here is retained (Fig 7B). As the two TIRs in a dsDNA circular transposon intermediate are covalently linked and in each other's immediate neighborhood, it may not be necessary to bind both in the same manner. If the first is bound as a donor TIR as seen here for R-TIR26, by analogy to the recipient TIR in the SCA complex, then either HTH$_B$ or NBD$_B$ might suffice to capture and bind the second TIR. It has been reported that junction integration by the IS30 transposase, unlike Figure-eight formation, does not require two full-length transposon ends and that one full-length TIR will compensate for truncation of the other (Szabó et al, 2010). ISCth4 TnpA forms dimeric complexes with TIR junction oligonucleotides in 1:2 DNA-to-protein stoichiometry, suggesting that only one TIR is bound initially in the absence of target supercoiled DNA. We also confirmed that the TIR junction mimic exhibits similar integration activity as observed for pre-cleaved TIRs (Fig 2H). The importance of TIR junction bending is indicated by our observations that TIR junctions with polyC/G spacers are not integrated as well as those with polyA/T spacers (Fig 6G, lanes 5–10) and that the flexibility induced by a nick in R26(j6) L14nick-3′OH accelerates the incorporation of a truncated recipient L-TIR that lacks subterminal base pairs (Appendix Fig S6).

Therefore, the asymmetric dimer we have observed here could carry out the symmetric intermediate integration step because symmetrical binding of both TIRs may not be necessary since the covalent link between them means that recognition of one necessitates the presence of the second. However, the exact mechanism of this step remains unclear, particularly how both TIRs of a junction are integrated and supercoiled target recognized.

Curiously, Figure-eight formation and TIR junction integration by TnpA show opposite preferences for the topology of their target substrates, as circular intermediates were generated only from a linearized substrate but TIR junction mimics were efficiently integrated only into a supercoiled target (Figs 2B, and 3A and B). These observations suggest the intriguing possibility that transposition initiation and production of a Figure-eight intermediate might be restricted to the relaxed segments of bacterial genomic DNA that are found, for instance, associated with replication forks during DNA replication. This could also make host replication factors immediately available to process Figure-eight intermediates into their dsDNA circular form. It would be interesting to see whether our target preference observations are generalizable and if copy-out/paste-in transposition might broadly serve as an shuttle for genomic elements from host DNA into mobile vectors capable of horizontal genomic transfer.

The results here add to the accumulating evidence that ID structural variation is a key property of DNA transposases that allows the constant RNase H-like fold to adapt and control mechanistically distinct reactions. For those transposases that have them, α-helical IDs appear essential for the chemical reaction steps of cleavage and double-stranded break formation as they provide appropriately placed critical residues, one of which is always a highly conserved His (Yuan & Wessler, 2011; Hickman et al, 2018). IDs are clearly multifunctional as they can also be deployed to bind an essential cofactor (GTP for the P element), form part of the binding interface for a partner protein (RAG1/RAG2), control multimerization (Hermes), and/or contribute to target or flanking DNA binding (Transib, RAG1, P element, and Hermes). It appears that the ID domains are one of the keys to understand the existence of the bewildering array of different transposase mechanisms. In ISCth4 TnpA, we have structurally characterized what appears to be the "core" insertion domain consisting of an antiparallel α-helical bundle. All other known transposase α-helical ID domains are related to this core fold through additional insertion of residues between the first and second α-helix (Appendix Fig S5A). They also appear to be dynamic, as in ISCth4 TnpA, the insertion domain moves as the reaction proceeds, and opening and closing have been observed in Transib and RAG1 (Ru et al, 2018; Liu et al, 2019; Chen et al, 2020). They may play a central role in directing the path of various DNA segments at different steps. It will be very interesting to establish how other transposase families that lack an insertion domain coordinate copy-out/paste-in transposition.

ISs have been shown to move within genomes on a clinically relevant time scale in patients, and one of the most important consequences of their transposition from the perspective of antibiotic resistance is the introduction of promoter elements near host genes. For example, IS256 family members have been documented to transpose in patients with persistent or recurring Staphylococcus aureus bacteraemia (Giulieri et al, 2018). When S. aureus isolates from MRSA patients were compared during the course of antibiotic treatment, a high frequency of IS256 insertions was detected with

some of the new insertion sites corresponding to genomic regions that might have affected the course of infection such as insertions around the antibiotic operon or upstream of the walKR operon associated with vancomycin resistance (Monk *et al*, 2019). The structures here reveal how promoter-like sequences closely resembling "TTTACA" and "TATAAT" are embedded in the IS*Cth4* transposon ends. This suggests a way how copy-out/paste-in transposons can accommodate DNA segments that are shared among TEs that assemble a hybrid promoter amidst TIR segments that make them unique. In the case of IS*Cth4*, the solution is that the two promoter elements are located where there are either no protein–DNA contacts or only contacts to the phosphate backbone rather than to specific bases (i.e., −35 is from bp 6–11 on R-TIR and −10 is from bp 8–13 on L-TIR; Figs 1D and F, and 5C). Thus, from the perspective of specific transposon recognition, the two hexamer sequences are located in the least important regions of the TIRs as had been previously postulated (Lewis *et al*, 2001). Remarkably, IS*256* has solved the problem in a different way (Maki & Murakami, 1997; Prudhomme *et al*, 2002) by placing the −35 region precisely at the tip of its right end from bp 1–6 (the −10 region extends from bp 12–17). The structures here show that there are few specific protein–DNA contacts at the very end of the IS*Cth4* transposon and is another region where it might be possible to introduce a conserved DNA motif innocuously in other related transposons. This strategy is used by the IS*2* element but with the other permutation as the −10 region is located at the tip of its left end (Lewis *et al*, 2004).

As revealed by the structures here, IS*Cth4* has served as a keyhole that has provided an opportunity to characterize several aspects of the elusive copy-out/paste-in transposition pathway. The IS*Cth4* transposase exhibits all of the important functions that are necessary for copy-out/paste-in transposition, and provides a solid foundation for future research of this important transposition pathway.

# Materials and Methods

### Expression and purification of TnpA

DNA encoding *E. coli* codon-optimized Tnp of IS*Cth4* from *C. thermocellum* (ATCC 27405) with an N-terminal thioredoxin fusion, histidine tag, and TEV cleavage site (TRX-TnpA) was purchased as a gBlock fragment (gBlock 1; Appendix Table S1) and ligated into pBAD/myc-His (Invitrogen). Transformed *E. coli* Top10 cells (Thermo Fisher Scientific) were grown in 2 l of LB media at 37°C to OD$_{600}$ ~ 0.8, and TRX-TnpA expression was induced with arabinose (Millipore Sigma) at a final concentration of 0.012% (*w/v*) for 18 h at 16°C. Cells were resuspended in 100 ml lysis buffer (25 mM HEPES pH 7.5, 0.5 M NaCl, 10 mM imidazole, 0.4 mM MgCl$_2$, ~ 2,000 U DNAse I (Roche), two tablets of cOmplete-ULTRA protease inhibitors (Roche), 5 mM β-mercaptoethanol) and lyzed with sonication. The lysate was centrifuged at 20,000 RPM (JA-20 rotor, Beckman Coulter) for 35 min at 4°C and filtered through a 0.45-μm syringe filter. Soluble TRX-TnpA was purified by nickel affinity chromatography on a His-trap column (GE Healthcare). Fractions containing TRX-TnpA were combined and incubated overnight with 0.5 mg of TEV protease at 4°C, then dialyzed against SP loading buffer (25 mM HEPES pH 7.5, 0.1 M NaCl, 5 mM β-mercaptoethanol). Cleaved TnpA was further purified with cation exchange

chromatography using a SP sepharose column (GE Healthcare) and gradient NaCl from 0.1 to 1 M. Fractions containing TnpA were concentrated to 5 ml and subjected to gel filtration using a Superdex 200 16/60 column and running buffer containing 25 mM HEPES pH 7.5, 0.5 M NaCl, 2 mM TCEP. Protein purity and integrity were monitored with SDS–PAGE. Protein was concentrated to 10–20 mg/ml and used immediately for crystallography or flash-frozen in liquid nitrogen. The active site mutant D175A was prepared in an identical manner (gBlock 2, Appendix Table S1).

### Expression of Se-methionine labeled TnpA

DNA encoding TRX-TnpA in the pBAD/myc-His vector was amplified with PCR using primers 1 and 2 and ligated into pst39 (Addgene #64009; Tan, 2001). Transformed *E. coli* B834(DE3) cells (Agilent) were grown in 2 l of SelenoMethionine Complete Medium (Molecular Dimensions) at 37°C to OD$_{600}$ ~ 0.8 and labeled TRX-TnpA expressed upon induction with 2 mM isopropyl β-D-1-thiogalactopyranoside for 18 h at 16°C and purified as described for unlabeled TnpA.

### Analytical size-exclusion chromatography

To assess oligomeric state and DNA-binding capacity, purified TnpA (typically at 40 μM or as indicated) was mixed with oligonucleotides in the indicated molar ratios and dialyzed overnight at 4°C into the running buffer (10 mM HEPES pH 7.5, 0.3 mM TCEP, and 100 mM NaCl). 10 μl was injected onto a Superose 6 Increase 3.2/30 column (GE Healthcare) calibrated with molecular weight standards (Sigma Aldrich) at 4°C with a flowrate of 50 μl/min.

### Sedimentation velocity analytical ultracentrifugation

Sedimentation velocity analytical ultracentrifugation was performed on a ProteomeLab™ XL-I or Optima XL-A analytical ultracentrifuge with an An-50 Ti rotor (Beckman Coulter) following standard protocols (Zhao *et al*, 2013). Sedimentation velocity experiments were conducted using 12- or 3-mm optical path length, two-channel charcoal-filled Epon centerpiece cells (Beckman Coulter) at 20°C and 50,000 RPM. Samples were prepared by dilution of protein and DNA stock solutions into the analysis buffer such that the final concentration of the buffer components was 10 mM HEPES pH 7.5, 100 mM, and 0.3 mM TCEP. The protein concentrations, protein–DNA molar ratios, and detection wavelengths for each experiment are indicated in the figures. Data were analyzed in SEDFIT 16.1c (Schuck, 2000) in terms of a *c(s)* distribution of sedimenting species with a resolution of 0.1 S, and a maximum entropy regularization of 0.68. The solution density, solution viscosity, and protein partial specific volume were calculated in SEDNTERP (Cole *et al*, 2008). A partial specific volume of 0.55 ml/g was used for DNA, and partial specific volumes for the nucleoprotein complexes were determined based on Traube's additivity principle. A bimodal *f/f$_0$ c(s)* distribution model was used to analyze sedimentation data for mixtures of DNA and nucleoprotein complexes to account for their different densities and the observed boundaries.

### Sedimentation equilibrium analytical ultracentrifugation

Sedimentation equilibrium analytical ultracentrifugation was performed on an Optima XL-A analytical ultracentrifuge with an

An-50 Ti rotor (Beckman Coulter). Experiments were conducted using 12-mm optical path length, two-channel charcoal-filled Epon centerpiece cells (Beckman Coulter), at 20°C and three rotor speeds of 6,000, 12,000, and 20,000 RPM. 5 μM protein stock solution was dialyzed into the analysis buffer for 24 h at 4°C. The slight precipitate that formed was removed by centrifugation. The remaining soluble TnpA (3 μM) was diluted with analysis buffer to 0.25 μM. Absorbance data collected at 230 nm were analyzed in terms of a single non-interacting species with mass conservation in SEDPHAT (Ghirlando, 2011).

### Normal mode analysis

Normal mode analysis of TnpA residues 160–406 was carried out using the iMODS web-server with the Cα coarse-graining model and the edNMA elastic network model according to the developer instructions (Orellana *et al*, 2010; López-Blanco *et al*, 2014).

### Circularization strand transfer assay *in vitro*

To assess targeted strand transfer activity necessary for the formation of a circular intermediate *in vitro,* two substrate plasmids were designed. Plasmid pUC19L was prepared by ligation of a gBlock fragment containing 35-bp L-TIR of IS*Cth4* (gBlock 3; Appendix Table S1) into NdeI/HindIII sites of pUC19 (Invitrogen). Plasmid pUC19LR was created by the QuikChange method using primers 3 and 4 to introduce 35-bp R-TIR of IS*Cth4* with the opposite orientation into another site of pUC19L (at base pair 1,618 between AmpR and ori segments), thereby creating a ~ 1,300-bp long segment between the TIRs. Both plasmids were verified with sequencing. pUC19LR and pUC19L were linearized with ScaI (NEB) before use in the indicated experiments.

Activity was assessed by mixing purified TnpA (final concentration 500 nM) with reaction buffer (10 mM Tris-acetate pH 7.9, 25 mM potassium acetate, 50 mM NaCl, 5 mM magnesium acetate, 50 μg/ml BSA, 5 mM DTT) and 500 ng of substrate plasmid pUC19LR or pUC19L in a final volume of 100 μl. The reaction was carried out at 37°C for 18 h and stopped by incubation with ~ 0.8 U proteinase K (NEB) for 1 h at 37°C and subsequent heating to 90°C for 30 min. A portion of the reaction mixture (5 μl) was subjected to PCR (95°C 20 s, 50°C 5 s, 72°C 10 s, 35 cycles) with primers *det1* and *det2*. For sequencing, linear product DNA was isolated using a 1.5% agarose gel run at 100 V for 45 min, visualized with GelRed (Biotium), extracted, blunt-cloned into pTZ57R/T vector according to the manufacturer's instructions (Thermo Fisher Scientific), and transformed into *E. coli* DH5α (NEB). For each experiment, 24 colonies (from total ~ 100) were selected with blue/white screening and sequenced.

The assay for formation of the Figure-eight intermediate has been replicated three times, and a representative result of one experiment is shown; contrast and brightness of gel images have been adjusted to make bands clearly visible.

### DNA integration assay *in vitro* and TSD identification

Qualitative assays were conducted in a total volume of 100 μl with the same reaction buffer as in the circularization assay using 2 μM DNA oligonucleotides, 0.25 μM TnpA, and 200 ng of pUC19 (Invitrogen) as the random DNA target. Reactions were incubated for 1, 4, and 24 h at 37°C (or as indicated in the figures where the gradient shows increase in time) and stopped with addition of 0.8 U of proteinase K (NEB) and 5 μl of 0.5 M EDTA. After 30-min incubation at 37°C and subsequent addition of glycogen, DNA was ethanol-precipitated, air-dried, and dissolved in water. DNA was visualized either with ethidium bromide or using a fluorescence imager Typhoon FLA7000 (GE Healthcare) on a 1.2% agarose gel run at 100 V for 70 min. To identify TSDs, linearized reaction products were extracted, subjected to two cycles of PCR with Pfu Ultra II polymerase (Agilent) to fill TSD gaps, ligated into the vector pCR-Blunt (Thermo Fisher), and transformed into *E. coli* DH5α.

To help the TIR junction mimic DNA to anneal properly, we typically added nine different non-transposon base pairs at both ends. The integration assay has been replicated with each oligonucleotide at least three times, and a representative result is shown. Contrast and brightness of gel images were adjusted to make bands clearly visible before cropping. For clarity, only areas of interest on gels are shown. Irrelevant lanes have been excluded.

### Crystallization of TnpA complexes

The protein:DNA complex between TnpA and R-TIR26 was assembled in 1:1 molar ratio, dialyzed overnight at 4°C against crystallization buffer (25 mM HEPES pH 7.5, 150 mM NaCl, 2 mM TCEP), and directly used in hanging drop vapor diffusion crystallization experiments. Crystals were obtained when 4 μl of protein solution (5.3 mg/ml) was mixed with 2 μl of precipitant consisting of 0.1 M HEPES pH 7.5 (Hampton), 150 mM NaCl (Quality Biochemical), 100 mM sodium acetate (Hampton), and 6% PEG4000 (Hampton) on a glass cover slip. Crystals grew over 7 days at 20°C to final size of ~ 0.5 × 0.5 × 0.5 mm and were flash-frozen in liquid nitrogen after gradual soaking in the precipitant solution mixed with 5, 12.5, and 25% glycerol mixtures with precipitant solution and overnight incubation at 20°C in 25% glycerol.

The Se-methionine-labeled complex of TnpA and R-TIR26 (PCC) was crystallized by mixing 3 μl of complex solution (6.2 mg/ml) in the same crystallization buffer as used for the native with 3 μl of precipitant 0.1 M HEPES pH 7.5, 150 mM NaCl, 100 mM sodium acetate, and 9% PEG4000. Crystals grew over 5 days to a final size of ~ 0.5 × 0.5 × 1 mm and were dehydrated in PEG4000 as follows. Crystals were transferred into a mixture of precipitant solution and 2.5% PEG4000 and incubated 20 min. The same procedure was repeated with increasing concentrations of PEG4000 5, 10, 15, 20, and 25% mixtures with precipitant solution. Crystals were incubated in 30% PEG4000 overnight at 20°C and flash-frozen in liquid nitrogen.

For the crystallization of the complex between TnpA and fR-TIR (PRC), the complex was assembled in 1:2 DNA:protein molar ratio, dialyzed against crystallization buffer (10 mM HEPES pH 7.5, 100 mM NaCl, 2 mM TCEP), and crystallized in a drop consisting of 3 μl of complex solution (7.9 mg/ml) and 3 μl of precipitant solution (0.1 M sodium cacodylate pH 6.5 (Hampton), 10 mM CaCl$_2$ (Hampton), 200 mM ammonium acetate (Analytical Reagents), 7% PEG4000 (Hampton)). Crystals grew over 12 days to final size ~ 0.2 × 0.2 × 0.2 mm and were subjected to dehydration as described above.

The complex between TnpA and RE26(j6)L14-5′rec (STC1) was assembled in 1:2 DNA:protein molar ratio, dialyzed against

crystallization buffer (10 mM HEPES pH 7.5, 100 mM NaCl, 2 mM TCEP), and crystallized in a drop consisting of 6 µl complex solution (2.4 mg/ml), 3 µl of precipitant (0.1 M sodium acetate pH 4.6 (Hampton), 0.2 M ammonium sulfate (Hampton), 0.25% PEG4000), and 0.9 ul of 0.1 M ZnCl$_2$ (Hampton). Crystals grew over 3 days to a final size 0.3 × 0.3 × 0.3 mm and were flash-frozen in liquid nitrogen after gradual soaking in 5, 12.5, and 25% glycerol mixtures with precipitant solution.

**Data collection and structure determination**

Diffraction data (Table 1) were either collected at the Advanced Photon Source, beamline ID-22, operated by SER-CAT or using CuKα radiation from a Rigaku 007HF source. Data were integrated and scaled internally in XDS and XSCALE (Kabsch, 2010). The Se-methionine labeled PCC was used for MAD experimental phasing at the wavelengths indicated in Table 1. Eighteen Se positions (from total 20) were identified using Hyss (Grosse-Kunstleve & Adams, 2003) and refined in SHARP (de la Fortelle & Bricogne, 1997) which was also used to calculate an initial solvent-flattened 4 Å experimental map (Bricogne *et al*, 2003) and was manually interpreted in O (Jones *et al*, 1991). Due to the limited resolution, the register of the sequence relative to the density could not be unambiguously established despite the known Se positions. The model was converted into polyserine chains and was optimized using the low-resolution refinement protocols available in CNS1.3 (Brünger *et al*, 1998). The sequence was then rebuilt and energy minimized to remove clashes. This model was subjected to extensive automatic rebuilding and refinement using density tools available in Rosetta 2017.36 (Bender *et al*, 2016). 300 decoys were generated, and a final model was built by the consensus of the five lowest energy ones while taking the Se positions into consideration. The model was further refined with Phenix (Adams *et al*, 2010) and verified using composite-simulated annealed omit maps. The model was also used as a search model in a successful molecular replacement calculation to determine the structure of the native PCC complex using phenix.-phaser (McCoy *et al*, 2007). The generated 3.5 Å 2m$F_o$–D$F_c$ map was used for further model rebuilding in COOT (Emsley *et al*, 2010). The resulting model was refined in Buster (Bricogne *et al*, 2017) and Phenix. The final structure contains 26 bp of R-TIR DNA and a dimer of TnpA showing 6–406 main chain residues (out of a total 407). The segment between G56 and D108 of protomer B is disordered. The structure of native PCC was used as a search model in the subsequent structure determinations of PRC and STC1 with molecular replacement in phenix.phaser (McCoy *et al*, 2007). Structures were manually rebuilt in COOT and refined in Phenix and Buster. All structures were verified with composite-simulated annealed omit maps computed in Phenix. The PRC structure lacked electron density for the terminal base pair in the flanking sequence which was not modeled. The relatively high average B-factors, while broadly consistent with the Wilson B-factors obtained from data scaling, are also consistent with the high solvent content of the crystals (71, 61, 70%) for the PCC, PRC, and STC1 structures, respectively. In the PCC and PRC complexes, B-factors of the CDs are higher than in N-terminal domains (PCC: CD$_A$ 166 Å$^2$, CD$_B$ 190 Å$^2$, N-term 146 Å$^2$; PRC: CD$_A$ 159 Å$^2$, CD$_B$ 168 Å$^2$, N-term 147 Å$^2$) while in STC1 the trend is the opposite (STC1: CD$_A$ 119 Å$^2$, CD$_B$ 148 Å$^2$, N-term 181 Å$^2$) reflecting the mobility of the CDs in the

PCC and PRC and the additional stabilization through the new interface between IDs in the STC1 structure.

## Data availability

The datasets related to the determination of X-ray structures produced in this study are available in the Protein Data Bank (http://www.wwpdb.org) under accession codes 6XG8, 6XGW, 6XGX.

**Expanded View** for this article is available online.

## Acknowledgements
We thank Mick Chandler for the suggestions of different transposases for screening and for his critical review of the manuscript. This work was supported by the Intramural Program of the National Institute of Diabetes and Digestive and Kidney Diseases, National Institutes of Health. Data were collected at Southeast Regional Collaborative Access Team (SER-CAT) 22-ID beamline at the Advanced Photon Source, Argonne National Laboratory. SER-CAT is supported by its member institutions (www.ser-cat.org/members.html), and equipment grants (S10_RR25528 and S10_RR028976) from the National Institutes of Health. The use of the Advanced Photon Source was supported by the U. S. Department of Energy, Office of Science, Office of Basic Energy Sciences, under Contract No. W-31-109-Eng-38. Some of the computations were carried out using the High Performance Computing Systems at the NIH.

## Author contributions
DK performed most of experiments in the study, crystallized, and determined structures of all presented complexes. RG performed AUC analysis. ABH performed initial activity and crystallization experiments. DK, ABH, and FD wrote the initial draft of the manuscript. SH screened proteins for solubility. FD supervised the study, and all authors participated in manuscript preparation and revision.

## Conflict of interest
The authors declare that they have no conflict of interest.

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
