## [Review Process File · The EMBO Journal]

Structures of ISCth4 transpososomes reveal the role of asymmetry in copy-out/paste-in DNA transposition

Dalibor Košek, Alison Hickman, Rodolfo Ghirlando, Susu He, and Fred Dyda
DOI: [10.15252/embj.2020105666](https://doi.org/10.15252/embj.2020105666)

Corresponding author(s): Fred Dyda (Fred.Dyda@nih.gov)

Review Timeline:	Submission Date:	19th May 20
	Editorial Decision:	7th Jul 20
	Revision Received:	7th Aug 20
	Accepted:	10th Sep 20

Editor: Hartmut Vodermaier

Transaction Report:

Thank you again for submitting your manuscript on copy-out/paste-in transposition structures for our editorial consideration. I have now heard back from three expert referees, whose reports are copied below for your information. As you will see, all of them consider your results interesting and well-presented, but they also raise a variety of specific queries that would need to be clarified prior to publication. Should you be able to satisfactorily address these points, we would be happy to consider a revised manuscript further for The EMBO Journal publication. Please do remember that it is our policy to allow only a single round of major revision, making it important to carefully answer to all referee points at this stage.

Generally, the particular suggestions and concerns are well-explained in the three reports and would mostly appear straightforward to address. Nevertheless, since I realize the difficulties many scientist experience with lab access and experimental work in the present COVID-19 pandemic situation, I would be open to discussing directly with you (via email exchange or call) options for how and within what timeline to best revise this study, once you may have had a chance to carefully consider the referee reports together with your coworkers. Our 'scooping protection' (meaning that competing work appearing elsewhere in the meantime will not affect our considerations of your study) would of course remain valid also during a potential extension of the revision period.

Further information on preparing and uploading a revised manuscript can be found below and in our Guide to Authors. Thank you again for the opportunity to consider this work for The EMBO Journal, and I look forward to hearing from you.

REFeree REPORTS

Referee #1:

This paper reports biochemical and molecular-level analysis of a bacterial copy-out-paste-in transposon mechanism. It reports three limited resolution (~3.5 Å) structures of three states along the transposition pathway: the pre-reaction state with flanking DNA (PRC); the active cleaved state (PCC); and the product of strand transfer from one transposon end into the recipient end (STC1). Together with in vitro analysis of reaction steps, the data provide fascinating insight into the molecular details of a previously poorly characterised transposition mechanism. I found the biochemical experimental data robust and the novel findings were put in the context of the earlier literature in an excellent and comprehensive discussion. This work is important and relevant to

understanding the emergence of antibiotic resistance, so the work will be of general appeal.

Major points:

(1) Likely because of the limited resolution of the data, it appears to have taken significant effort, skill and determination to solve and then refine the crystal structures, first using selenomet-MAD analysis of 4 Å PCC crystals and molecular replacement. However, I am concerned about the high overall B-factors for each of the structures: 151-169 Å². Could the authors suggest an explanation for these high values? Are particular domains more mobile than others, or are the B-factors high across all atoms? I would like to see a figure included (supplementary) showing the quality of the electron density of the composite simulated annealed omit maps that were computed with Phenix.

(2) Comparison of the structures of each state reveals the movement of the domains and of the transposon DNA, flanking DNA and linker DNA as the transposition reaction proceeds. These structures are represented mostly as ribbon diagrams. What I was missing was some close-up views of the positions of each of the active sites, showing the atomic positions of the reactive ends of the DNA with respect to the active site amino acids. The authors discuss the "formation of catalytically functional units" so this is important. There appears to be some evidence in the supplementary movies (which are very useful for demonstrating the domain movements and asymmetry of this unusual transposition reaction) of the active site amino acid positions changing between the different trapped states, so this is potentially very interesting. In the PRC complex the 3'-OH is held away from the active site, which makes sense. But are the catalytic amino acids in an active or inactive conformation in this PRC structure? Later the authors state that "in the PCC (fig 4B) the 3'-OH of the TS points toward the active site, a position consistent with donor TIR binding"; but is it consistent with catalysis of the expected reaction?

(3) None of the structures contain metal ions in the active sites, but these ions are presumably required for catalysis. [Metal ions were included in the activity assays, but complexes were equilibrated against buffer that excludes metals for crystallisation and the crystal conditions do not include Metal ions]. "The active site residues (D175, D241, E348) are in a shallow cavity and converge towards the free 3'-OH of R-TIR26 (Fig 4E)". But are these amino acids in the correct conformation for metal binding (and catalysis)? (And is this referring to the PCC structure?)

(4) For each of the structures, could the authors please provide a figure showing the atomic positions of the active site residues, the electron density and the positions of the scissile DNA phosphates, indicating whether or not they are positioned so that catalysis can occur and if the metal ions required for catalysis could bind.

Other minor points:

(5) Did they consider if there is any difference in affinity for the binding of TnpA to L-TIR compared to R-TIR? And if so, could this be relevant to the mechanism?

(6) In STC1 they have captured the product of the junction formation....." Due to the asymmetry imposed by binding to the donor TIR, flanking DNA rather than the transposon tip of the recipient TIR would be directed into the active site of CDB. This provides an elegant explanation for the strand transfer offset that is at the heart of circle junction formation." What is strand transfer offset?

(7) On Page 15 it is stated "One of the most important roles of the cdA binding site is to direct the

tip of the transposon into the active site of CDB" . But is it not to position accurately the tip in the active site for catalysis?

(8) Include a space between the value and unit throughout.

The extensive figures and two movies are generally nicely presented but I have some suggestions for improvements to aid clarity:

9. It would be useful to add the complex labels (PRC, PCC, STC1) in Fig 1C and to indicate the 'Figure 8' intermediate discussed in the text.
10. The domain colour scheme panel in Figure 7C would be more useful positioned earlier on, in Figure 1 perhaps.
11. In my copies of the diagrams the colours in the figures did not match those in the nearby schematics, so perhaps the dropper facility could be used to prepare schematics.
12. In Fig 1B, colour the target DNA differently to the donor flanking DNA - both are black - to make it easier to understand the co-integrate intermediate. And retain that colour scheme in the structural figures. [It was not altogether clear to this reader how replication and resolution produces the regenerated donor, but perhaps that is not important here]. The position of the R-TIR arrow in this panel suggests the same strand is cleaved at both ends, but it would be the lower strand that is cleaved at the R-TIR as represented here.
13. To avoid confusion the 5' and 3' ends of the transposon DNA should be indicated. The top sequence in Fig1E appears to be a R-ITR- spacer-L-ITR junction as written (is this a 2-fold rotation compared to the schematic in D?), whereas L-TIR-spacer-R-TIR junction sequences appear in Fig 2C.
14. Be consistent with naming the middle DA substrate: R35r41 in the Fig 2 legend, but R35(j6)r35 on the figure. Similarly, in Fig. 3A.
15. Move gel lane numbers to below the gel and specify what the gradient and the numbers above the gradient symbols indicate. In Fig 2G there are 26 lanes, not 27 as labelled.
16. The source data file for Fig 2E and 3A shows the formation of a slowly migrating band when the TIR is 25 bp or more - How do they think this band arises?
17. Figure 5 is a very nice figure. The nomenclature NTS and TS could be introduced earlier. Colour C26' red throughout for consistency. Explain the nomenclature cdA and ntA - the binding sites on the TIR of CDB and NDBA respectively - in the figure legend.
18. State explicitly which structure is shown in each figure panel.
19. In figure 6E, is there a base - C15'- missing? (G33 at junction is unpaired). Was this the base removed to aid crystallisation? Indicate in 6F if this is the case.

Referee #2:

This manuscript examines the biochemical and structural properties of the transposase (TnpA) from the copy-out/paste-in TE IS_Cth4. This class of TEs is relevant medically because of the ability to mediate antibiotic resistance and is not well understood mechanistically and no structural information existed previously. The biochemical data demonstrate that purified TnpA is active for Figure 8 formation (the initial step of the transposition reaction) as well as insertion into target DNA, provide information on the TIR requirements for these reactions, and characterize the association state of TnpA (indicating a concentration-dependent monomer-dimer equilibrium in the absence of DNA and binding of the donor TIR exclusively as a dimer). The biochemical analyses are solid and for the most part clearly presented. They provide useful mechanistic insights and important

information for interpretation of the structures.

Three different structures, arising from x-ray crystallography, are presented, the pre-reaction complex (PRC), the pre-cleaved complex (PCC), and the strand transfer complex 1 (STC1), which the authors argue represents an approximation of the Figure 8 complex. From these structures, a wealth of interesting insights emerge. Most intriguing is the intrinsic asymmetry of the TnpA dimer in all three complexes, caused by a distinctive set of interactions involving the dimerization domain (DD), N-terminal DNA binding domain (NDB), and the helix-turn-helix domain (HTH). Strikingly, in one TnpA subunit, the NDB is unstructured and not resolved in the structures, with the consequence that the TnpA dimer has the ability to interact fully with only one TIR. This nicely reflects the initial reaction steps for this class of transposases wherein one TIR is the donor and the other the recipient of strand transfer and hence play quite distinct roles. The structures also reveal the interactions that mediate dimerization and DNA binding, conformational changes that likely underlie movement of the scissile phosphate into close proximity of the active site, and an interesting movement of the catalytic domain that leads TnpA to close over the junction spacer DNA, requiring a large 120 degree bend in the DNA at the junction. Overall, the study provides the first structural information for this class of TEs and substantial new insight into the mechanism of transposase action. Interesting ideas are developed in the Discussion, which highlights some important future questions regarding copy-out/paste-in transposases.

This is an excellent study that is well suited for publication in EMBO J. It could be improved by addressing a few minor points:

1. As indicated in Fig. 4C, the DNA substrate for formation of STC1 contains a nick/gap on one strand, and the figure legend notes that this was essential for obtaining diffracting crystals. Presumably this relates to the need to create a 120 degree bend in the DNA-this interruption of one strand would greatly relieve stress on the DNA caused by bending. I suggest that the authors make this connection explicitly on page 16 where the bend is first introduced.
2. Figure 2B: Since product detection is by PCR, it is not possible to tell how efficient the reaction is. It would be helpful for the authors to indicate approximate reaction efficiency if that can be estimated. It is not until the Discussion that the authors note that Figure 8 product formation is very inefficient.
3. Figure 2H is not mentioned in the text.
4. Fig. 3A: why is any linear product at all generated in lanes 4-6 with one TIR completely scrambled? One would expect only single end events that would yield relaxed circles, as in lanes 13-15. And why are relaxed circles not seen in lanes 4-6? Do these data undermine the following statement (bottom of page 8, top of page 9)? "The absence of linearized target plasmid indicates that when only one end of a junction can be integrated, TnpA does not utilize another junction substrate from the reaction mixture."
5. Also Fig. 3A: Regarding the 5'OH substrate, the authors write "Both TIRs were integrated into supercoiled DNA (lanes 10-12) but with reduced efficiency relative to R35(j6)L35 nick-3'OH." This is true at 24 hr but not at 1 or 4 hours. In fact, at 1 hour, the 5'OH substrate is more efficient. Why? This should be addressed in the manuscript.
6. Page 17: "We also observed an independent synergetic effect of base pairs 1-14 and 15-26 of the L-TIR...". I don't think "synergistic" is the correct term here. The data indicate that both regions of the L-TIR contribute to activity, but no synergism (greater activity when both are present than would be predicted by the activity when one or the other is present) is evident. In fact, when both are present, the increase is modest over just one or the other.

Referee #3:

The manuscript concerns an important and underserved member of the diverse family of DDE DNA transposons. Unlike the commonly discussed DDE elements where all of the chemistry occurs within the individual ends of the element, this group makes its initial breaking and joining event to the other "end" of the element. Host initiated DNA replication produces the second strand of the element at the donor site and the second strand of the element that makes the donor molecule that is integrated at the target via simple insertion event (hence the name copy-out and paste-in). One of the many intriguing molecular gymnastics that needs to occur involves an asymmetric event, something not explained in the "textbook" model of transposition. In this paper they establish a reconstituted in vitro reaction with purified proteins with an element found in multiple copies in a strain of *Clostridium thermocellum* and use these proteins for structural work. The major advances in the paper are discovering how the initial symmetric complex can form and the coordinating role of an insertional domain.

The manuscript does a nice job explaining this complicated system and why it is important. They also do a nice job walking the reader through their logic. Not every aspect of the experiments is perfectly tidy, but these do not invalidate their conclusions. There are a few points that do need to be addressed as explained below.

The idea that "...Figure 8 intermediate prefers relaxed DNA as is characteristic for segments of prokaryotic genomic DNA" is not clear (Page 22). There should be periods of relaxed supercoiling within the genome with organisms across all domains of life, not specifically prokaryotes. Maybe they are referring to negative supercoiling? (something not addressed in their work?) Without an extensive and believable explanation for what they mean I think this idea should be dropped. One idea to consider could be a link with DNA replication. DNA would be relaxed behind the DNA replication fork and could probably be suggested as a cue for initiating Figure-8 formation. This could allow replication factors to be more readily available to initiate replication of the second strand of each copy of the element and help displace the circle. If plasmids are expected to be supercoiled it could support the idea of favoring transposition to molecules that could facilitate horizontal transfer.

Optimal growth temperature for *Clostridium thermocellum* is 55C, but all of the experiments were done at 37C. Were normal growth temperatures for *C. thermocellum* tested in vitro? This could have some effect on the interpretation of supercoiling and AT bias and the complications with protein aggregation.

The paper just seems to fall off at the end without completing the thought at the end of the discussion?

In Figure 1B and 1C as shown, the donor and target DNAs are in opposite 5 prime to 3 prime orientations. Because each is turned on its side, it is hard to say which should be which, but for clarity the same convention should be used relative to each other.

Figure 1C would be better if the points of joining to the target were more clearly at the edges of what will become the TSD.

Figure 2A and D would be much clearer if it was shown with both DNA strands.

Figure 7 - Why was the NDB domain left out of the "B" protomer? Was this because it lacked resolution in the structure, maybe show with dashed line instead?

Page 3, paragraph 1 = "Many ISs also contain regulatory elements such as promoters..." Clarify that talking about promoters that will transcribe the region outside the element upon integration.

Page 3, last line, paragraph 1 - Not all of these references given involve CRISPR-Cas9 and "efficiency" is vague. Should state something about guide-RNA targeting generally with CRISPR-Cas. Guide RNA targeting with Cas9 is a new function of the system so how can it be an improvement in "efficiency"?

Page 4, last paragraph - "...which drives transposase expression..." should say something along the lines of "...drives increased transposase expression..." because the transposase must have been expressed at some level to allow excision of the element to make the circular form.

Page 6, first paragraph - Write out *Clostridium thermocellum*.

Point by point responses:

Referee #1:

This paper reports biochemical and molecular-level analysis of a bacterial copy-out-paste-in transposon mechanism. It reports three limited resolution (~ 3.5 Å) structures of three states along the transposition pathway: the pre-reaction state with flanking DNA (PRC); the active cleaved state (PCC); and the product of strand transfer from one transposon end into the recipient end (STC1). Together with in vitro analysis of reaction steps, the data provide fascinating insight into the molecular details of a previously poorly characterised transposition mechanism. I found the biochemical experimental data robust and the novel findings were put in the context of the earlier literature in an excellent and comprehensive discussion. This work is important and relevant to understanding the emergence of antibiotic resistance, so the work will be of general appeal.

Major points:

(1) Likely because of the limited resolution of the data, it appears to have taken significant effort, skill and determination to solve and then refine the crystal structures, first using selenomet-MAD analysis of 4 Å PCC crystals and molecular replacement. However, I am concerned about the high overall B-factors for each of the structures: 151-169 Å². Could the authors suggest an explanation for these high values? Are particular domains more mobile than others, or are the B-factors high across all atoms? I would like to see a figure included (supplementary) showing the quality of the electron density of the composite simulated annealed omit maps that were computed with Phenix.

The referee is correct that the Phenix-optimized B factors are high. However, this is not unusual for protein/nucleic acid complexes that diffract to a limited resolution. One reason is the quality of the crystal lattice that is held together by DNA/DNA or protein/DNA interactions that are often not as solid as protein/protein interactions. Furthermore, the solvent content of the lattices reported in the paper are relatively high. We have been conservative in defining the high resolution limit of the data used in refinement, precisely because including weak and incomplete resolution shells would have raised the refined B factors to even higher values. For the PCC data, we cut the resolution included at 3.5 Å where the overall $I/\sigma(I)$ was at 2.1 and the Wilson B factor from XDS was 135 Å²; for the PRC the same numbers are 2.7 $I/\sigma(I)$ at 3.5 Å and 119 Å²; and for the STC1 structures are 1.71 $I/\sigma(I)$ at 3.5 Å and 111 Å². The solvent contents of the three lattices are 71 %, 61 %, and 70 % respectively. The Wilson B factors are broadly consistent with the Phenix-refined values and therefore we conclude that they are reflecting the falloff of the measured intensities with increasing scattering angle, and are properties of the data and not the consequence of our interpretation and modeling of the electron density.

Among domains, there are only relatively small differences in overall B-factors. In the PCC and PRC, catalytic domains have generally a higher B-factor than the N-terminal domains (PCC: CD_A 166 Å², CD_B 190 Å², HTH+NDB+DD 146 Å²; PRC: CD_A 159 Å², CDB 168 Å², HTH+NDB+DD 147 Å²).

In STC1, the trend is the opposite with lower B-factors for the catalytic domains than the N-terminal domains (CD_A 119 Å², CD_B 148 Å², HTH+NDB+DD 181 Å²), consistent with the likely stabilization of catalytic domains due to the new interface between insertion domains. A short paragraph that discusses the B-factors has been added to the Materials and Methods section (p. 32).

We agree with the referee that given the limited resolution and the fact that the final structures were solved using molecular replacement, we should include views of the composite, simulated annealed omit maps, and these are now presented in the new Figure S7A, B and C.

(2) Comparison of the structures of each state reveals the movement of the domains and of the transposon DNA, flanking DNA and linker DNA as the transposition reaction proceeds. These structures are represented mostly as ribbon diagrams. What I was missing was some close-up views of the positions of each of the active sites, showing the atomic positions of the reactive ends of the DNA with respect to the active site amino acids. The authors discuss the "formation of catalytically functional units" so this is important. There appears to be some evidence in the supplementary movies (which are very useful for demonstrating the domain movements and asymmetry of this unusual transposition reaction) of the active site amino acid positions changing between the different trapped states, so this is potentially very interesting. In the PRC complex the 3'-OH is held away from the active site, which makes sense. But are the catalytic amino acids in an active or inactive conformation in this PRC structure? Later the authors state that "in the PCC (fig 4B) the 3'-OH of the TS points toward the active site, a position consistent with donor TIR binding"; but is it consistent with catalysis of the expected reaction?

Given the limited resolution of the structures, the absence of the scissile phosphate in the PCC structure, and the lack of bound metal ions, we would prefer not to discuss side chain orientations at the active site. However, the referee is correct that our description of what we observed at the active sites was not clear. To correct this and also considering the referee's comment (4) below, we have added Figures S7D, E and F to show the active site residues, $2F_o - F_c$ electron density maps, and scissile phosphates (either as present in the structure or modeled where it was not part of the crystallized construct). We have also added the following sentences to p.16-17:

"The comparison of the active site of PCC to the active site of Hermes when bound to DNA (Hickman et al, 2018) reveals that although the 3'-OH of the TIR end is in proximity to the catalytic residues, the scissile phosphate that would be present on the TIR end is still displaced from the equivalent position of the scissile phosphate in the Hermes complex by ~5 Å (assuming ideal geometry). This is likely due to a combination of factors including the lack of bound metal ion to organize the active site, stabilization of the terminal nucleotide by base pairing to the complementary strand, and the lack of flanking DNA that would be part of the authentic pre-cleaved state."

(3) None of the structures contain metal ions in the active sites, but these ions are presumably required for catalysis. [Metal ions were included in the activity assays, but complexes were equilibrated against buffer that excludes metals for crystallization and the crystal conditions do not include Metal ions]. "The active site residues (D175, D241, E348) are in a shallow cavity and

converge towards the free 3'-OH of R-TIR26 (Fig 4E)". But are these amino acids in the correct conformation for metal binding (and catalysis)? (And is this referring to the PCC structure?) Unfortunately, we were able to obtain diffracting crystals only in the absence of metal ions with the exception of the PRC structure where the crystallization was possible only after the addition of Ca^{2+} . Nevertheless, we did not observe electron density that we could reliably interpret as a bound Ca^{2+} in the PRC active site. As we indicated above, we have added Fig S7D, E, and F to show TnpA active site residues in comparison to the Hermes active site with metal bound (PDB code 6DWW) and we have added the following to the text (p. 14) to indicate that in none of the structures are the active sites correctly assembled in the absence of bound metal ions:

"Although the resolution of the structures is limited and none of them contains metal ions in the active sites, in the PCC and PRC, the side chains of D175 and D241 are in the appropriate conformation for metal binding and catalysis (Fig S7D and S7E) whereas the side chain of E348 is pointing away from the presumed metal binding site. In STC1, the active site is further disordered and the D175 side chain is turned in the opposite direction that observed in the PCC and PRC structures (Fig S7F)."

(4) For each of the structures, could the authors please provide a figure showing the atomic positions of the active site residues, the electron density and the positions of the scissile DNA phosphates, indicating whether or not they are positioned so that catalysis can occur and if the metal ions required for catalysis could bind.

As requested, three new figures have been added to address these points (Fig S7D,E, and F).

Other minor points:

(5) Did they consider if there is any difference in affinity for the binding of TnpA to L-TIR compared to R-TIR? And if so, could this be relevant to the mechanism?

In our initial biochemical experiments, we extensively investigated DNA binding with versions of both L-TIR and R-TIR separately, yet we did not observe any significant difference in binding affinity using SEC and we interpret this as consistent with the conserved protein binding sites along L- and R-TIRs. We believe it is important that we observed strand transfer from L-TIR to R-TIR and vice versa, indicating that they both can serve as a donor TIR as this is consistent with the previously reported properties of IS256 (Loessner et al, 2002; Prudhomme et al, 2002). We have added the following sentence to the Results section describing the SEC experiments: "We observed no difference in behavior when we used L-TIR substrates, suggesting that they have similar affinities and consistent with our observation that either end can serve as the donor TIR."

(6) In STC1 they have captured the product of the junction formation....." Due to the asymmetry imposed by binding to the donor TIR, flanking DNA rather than the transposon tip of the recipient TIR would be directed into the active site of CDB. This provides an elegant explanation for the strand transfer offset that is at the heart of circle junction formation." What is strand transfer offset?

We apologize, as we indeed failed to properly define this terminology. By offset, we meant the number of base pairs from the tip of recipient TIR to the site where the donor TIR is integrated.

To be clear, we have modified the relevant sentence to remove the "offset" terminology altogether:

"This provides an elegant explanation for strand transfer into flanking sequence with the concomitant generation of a spacer that is at the heart of circle junction formation."

(7) On Page 15 it is stated "One of the most important roles of the *cdA* binding site is to direct the tip of the transposon into the active site of CDB". But is it not to position accurately the tip in the active site for catalysis?

Although we have not captured the transposon tip in a position poised for catalysis, the referee is correct that the *cdA* binding site likely plays a role in both these aspects. We have therefore modified the sentence as follows:

"One of the most important roles of the *cdA* binding site is to direct the tip of the transposon into the active site of CD_B, and presumably to position it appropriately for catalysis (Fig 4D and 5A and 5B)."

(8) Include a space between the value and unit throughout.

We thank the referee for pointing this out and spaces are now included throughout the manuscript.

The extensive figures and two movies are generally nicely presented but I have some suggestions for improvements to aid clarity:

9. It would be useful to add the complex labels (PRC, PCC, STC1) in Fig 1C and to indicate the 'Figure 8' intermediate discussed in the text.

We have modified Fig 1C to label the Figure 8 intermediate. As for the complex labels, while it would be possible to add one for the PRC, the PCC and STC1 would be problematic: The PCC state is not explicitly represented in the figure and STC1 only approximates the Figure 8 intermediate as it does not include the flanking regions that are present on both transposon ends. Therefore, we would prefer not to add these labels to Fig 1C as we fear that their inclusion might cause confusion.

10. The domain colour scheme panel in Figure 7C would be more useful positioned earlier on, in Figure 1 perhaps.

We agree with the referee and we have added the color scheme to the DNA diagrams in Fig 1D, and the protein domain color scheme is now introduced in Fig 4, next to the presented structures. (Although the domain organization of the C-terminal region of TnpA was predicted before our structures were determined, it is only the structural results presented in Fig 4 that established the domain organization of the N-terminal region.)

11. In my copies of the diagrams the colours in the figures did not match those in the nearby schematics, so perhaps the dropper facility could be used to prepare schematics.

We agree with the referee and all colors in the schematics have now been adjusted to match the figures.

12. In Fig 1B, colour the target DNA differently to the donor flanking DNA - both are black - to make it easier to understand the co-integrate intermediate. And retain that colour scheme in the structural figures. [It was not altogether clear to this reader how replication and resolution produces the regenerated donor, but perhaps that is not important here]. The position of the R-TIR arrow in this panel suggests the same strand is cleaved at both ends, but it would be the lower strand that is cleaved at the R-TIR as represented here.

We agree with the referee and following his/her suggestions, we have changed the color of the target DNA from black to purple. We have also added a reference to the co-integrate intermediate resolution step to the figure legend although we feel that discussing it would be beyond the scope of this article. As suggested, the color scheme has been adjusted (transposon flanking sequences in magenta) in Fig 1 and retained across the manuscript. Only panels in Fig 6 and S7 have different colors to allow us to highlight the differences between two structures or to show the entire structure.

13. To avoid confusion the 5' and 3' ends of the transposon DNA should be indicated. The top sequence in Fig1E appears to be a R-ITR- spacer-L-ITR junction as written (is this a 2-fold rotation compared to the schematic in D?), whereas L-TIR-spacer-R-TIR junction sequences appear in Fig 2C.

As suggested by the referee, we have added 5' and 3' labels wherever possible including Fig 1E and Fig 1A. The referee is correct that we had inverted the orientations in Fig 1D and 2C; we apologize and these have now been standardized (the orientation in Fig 2C has been switched) and an explanation has been added to the Fig 2C legend.

14. Be consistent with naming the middle DA substrate: R35r41 in the Fig 2 legend, but R35(j6)r35 on the figure. Similarly, in Fig. 3A.

We thank the referee for noticing this, and the name has been standardized to R35r41 throughout.

15. Move gel lane numbers to below the gel and specify what the gradient and the numbers above the gradient symbols indicate. In Fig 2G there are 26 lanes, not 27 as labelled.

We thank the referee for pointing these out. We have moved the gel lane numbers in Fig 2B, Fig 6G, Fig S1B, and Fig S6. We have also corrected the lane numbering in Fig 2G, time points are now indicated next to gradient symbols, and an explanation has been added to the Materials and Methods section.

16. The source data file for Fig 2E and 3A shows the formation of a slowly migrating band when the TIR is 25 bp or more - How do they think this band arises?

Some preps of substrate plasmid pUC19 contained additional slow-migrating species between 4 and 5 kb (shown in Fig 2F lane 1). We assume that these are supercoiled dimers or catenanes of pUC19, since they produce a single band consistent with linear pUC19 when cleaved with a restriction enzyme (Fig 3B lane 1 vs lane 3, linearized with *ScaI*). We speculate that the band in Fig 2E the referee noticed is a single-stranded insertion of substrate oligonucleotide into these dimers, perhaps resulting in a supercoiled pUC19 joined to the relaxed one. In the source data

for Fig 3A, there are additional slower migrating bands (slower than 10kb, the upper band in the marker lane) in lanes 13-15. We believe these are the result of multiple integration events into the multiple forms of pUC19 in the prep.

17. Figure 5 is a very nice figure. The nomenclature NTS and TS could be introduced earlier. Colour C26' red throughout for consistency. Explain the nomenclature cdA and ntA - the binding sites on the TIR of CDB and NDBA respectively - in the figure legend.

Following the referee's suggestions, C 26' has now been indicated in red in all the relevant figures and we have added an explanation of the binding sites to the figure legend.

18. State explicitly which structure is shown in each figure panel.

We appreciate this suggestion and each panel is now so labeled.

19. In figure 6E, is there a base - C15'- missing? (G33 at junction is unpaired). Was this the base removed to aid crystallisation? Indicate in 6F if this is the case.

These were a consequence of a poor choice of figure orientation in this region. The view has been rotated slightly to fix this optical illusion, and we thank the referee for pointing this out.

Referee #2:

This manuscript examines the biochemical and structural properties of the transposase (TnpA) from the copy-out/paste-in TE IS_{Cth4}. This class of TEs is relevant medically because of the ability to mediate antibiotic resistance and is not well understood mechanistically and no structural information existed previously. The biochemical data demonstrate that purified TnpA is active for Figure 8 formation (the initial step of the transposition reaction) as well as insertion into target DNA, provide information on the TIR requirements for these reactions, and characterize the association state of TnpA (indicating a concentration-dependent monomer-dimer equilibrium in the absence of DNA and binding of the donor TIR exclusively as a dimer). The biochemical analyses are solid and for the most part clearly presented. They provide useful mechanistic insights and important information for interpretation of the structures.

Three different structures, arising from x-ray crystallography, are presented, the pre-reaction complex (PRC), the pre-cleaved complex (PCC), and the strand transfer complex 1 (STC1), which the authors argue represents an approximation of the Figure 8 complex. From these structures, a wealth of interesting insights emerge. Most intriguing is the intrinsic asymmetry of the TnpA dimer in all three complexes, caused by a distinctive set of interactions involving the dimerization domain (DD), N-terminal DNA binding domain (NDB), and the helix-turn-helix domain (HTH). Strikingly, in one TnpA subunit, the NDB is unstructured and not resolved in the structures, with the consequence that the TnpA dimer has the ability to interact fully with only one TIR. This nicely reflects the initial reaction steps for this class of transposases wherein one TIR is the donor and the other the recipient of strand transfer and hence play quite distinct roles. The structures also reveal the interactions that mediate dimerization and DNA binding, conformational changes that likely underlie movement of the scissile phosphate into close proximity of the active site, and an interesting movement of the catalytic domain that leads

TnpA to close over the junction spacer DNA, requiring a large 120 degree bend in the DNA at the junction. Overall, the study provides the first structural information for this class of TEs and substantial new insight into the mechanism of transposase action. Interesting ideas are developed in the Discussion, which highlights some important future questions regarding copy-out/paste-in transposases.

This is an excellent study that is well suited for publication in EMBO J. It could be improved by addressing a few minor points:

1. As indicated in Fig. 4C, the DNA substrate for formation of STC1 contains a nick/gap on one strand, and the figure legend notes that this was essential for obtaining diffracting crystals. Presumably this relates to the need to create a 120 degree bend in the DNA-this interruption of one strand would greatly relieve stress on the DNA caused by bending. I suggest that the authors make this connection explicitly on page 16 where the bend is first introduced.

We agree with the referee's suggestion, and we have moved the discussion of the STC1 substrate in the main text as follows: "We were able to capture this state crystallographically only after introducing a nick and a one nucleotide recess on the non-transferred strand of R-TIR (R26(j6)L14-5' rec). This likely relieved the stress on the DNA caused by bending, and helped to stabilize the complex."

2. Figure 2B: Since product detection is by PCR, it is not possible to tell how efficient the reaction is. It would be helpful for the authors to indicate approximate reaction efficiency if that can be estimated. It is not until the Discussion that the authors note that Figure 8 product formation is very inefficient.

As the referee suggested, we indeed do not think there is a good way to estimate the reaction efficiency given that we had to use PCR. At the same time, to better describe the situation, we have modified the following sentence in the Results section:

"Although we were unable to directly detect the strand transfer step that generates the Figure 8 intermediate using oligonucleotides, we recapitulated Figure 8 intermediate formation using a PCR-based *in vitro* strand transfer plasmid assay in which we introduced the 35-bp L-TIR and R-TIR sequences into pUC19 ("pUC19LR", Fig 2A) or the L-TIR alone ("pUC19L")."

3. Figure 2H is not mentioned in the text.

We thank the referee for bringing this to our attention - it has now been added (p. 8).

4. Fig. 3A: why is any linear product at all generated in lanes 4-6 with one TIR completely scrambled? One would expect only single end events that would yield relaxed circles, as in lanes 13-15. And why are relaxed circles not seen in lanes 4-6? Do these data undermine the following statement (bottom of page 8, top of page 9)? "The absence of linearized target plasmid indicates that when only one end of a junction can be integrated, TnpA does not utilize another junction substrate from the reaction mixture."

We thank the referee for pointing out this apparent inconsistency, and we apologize for not addressing this adequately. Our data suggest when two TIRs are joined in the form of the junction, if integration of one end is compromised then only the other TIR is integrated and a

second molecule is not bound or processed. However, if one TIR is flanked by random DNA (as in Fig 3A, lanes 4-6), then the situation resembles that of pre-cleaved TIRs, and two ends can be coordinately integrated. Thus, it appears that these two classes of substrates bind differently and are acted upon differently. We have rewritten the sections describing these reactions to explain.

5. Also Fig. 3A: Regarding the 5'OH substrate, the authors write "Both TIRs were integrated into supercoiled DNA (lanes 10-12) but with reduced efficiency relative to R35(j6)L35 nick-3'OH." This is true at 24 hr but not at 1 or 4 hours. In fact, at 1 hour, the 5'OH substrate is more efficient. Why? This should be addressed in the manuscript.

We agree with the referee that more discussion of this issue is needed; however, we would like to keep the interpretation of these time courses at a minimum as at this point we can only speculate. Whereas R35(j6)L35nick-3'OH represents a prenicked junction where the nick is situated at the site of expected initial cleavage and is an intermediate along the catalytic cycle, R35(j6)L35nick-5'OH places the nick on the opposite strand and does not represent a bonafide intermediate in the pathway of integration. One possibility is that, during cleavage and integration of R35(j6)L35nick-5'OH, the covalent link between TIRs might be broken before the reaction is complete with unclear consequences on integration efficiency and kinetics. We have modified the text as follows:

"Furthermore, the kinetics of the reaction were altered. While we could detect a product at an early time point with R35(j6)L35nick-5'OH, subsequent accumulation of product was much slower. It is possible that the reaction is compromised due to the double-strand break upon cleavage at the L-TIR."

6. Page 17: "We also observed an independent synergetic effect of base pairs 1-14 and 15-26 of the L-TIR...". I don't think "synergistic" is the correct term here. The data indicate that both regions of the L-TIR contribute to activity, but no synergism (greater activity when both are present than would be predicted by the activity when one or the other is present) is evident. In fact, when both are present, the increase is modest over just one or the other.

We agree with the reviewer and have changed the sentence to indicate that "bp 1-14 and 15-26 of the L-TIR independently contribute to the activity..."

Referee #3:

The manuscript concerns an import and underserved member of the diverse family of DDE DNA transposons. Unlike the commonly discussed DDE elements where all of the chemistry occurs within the individual ends of the element, this group makes its initial breaking and joining event to the other "end" of the element. Host initiated DNA replication produces the second strand of the element at the donor site and the second strand of the element that makes the donor molecule that is integrated at the target via simple insertion event (hence the name copy-out and paste-in). One of the many intriguing molecular gymnastic that needs to occur involves an asymmetric event, something not explained in the "textbook" model of transposition. In this paper they establish a reconstituted in vitro reaction with purified proteins with an element

found in multiple copies in a strain of *Clostridium thermocellum* and use these proteins for structural work. The major advances in the paper are discovering how the initial symmetric complex can form and the coordinating role of an insertional domain.

The manuscript does a nice job explaining this complicated system and why it is important. They also do a nice job walking the reader through their logic. Not every aspect of the experiments is perfectly tidy, but these do not invalidate their conclusions. There are a few points that do need to be addressed as explained below.

1. The idea that "...Figure 8 intermediate prefers relaxed DNA as is characteristic for segments of prokaryotic genomic DNA" is not clear (Page 22). There should be periods of relaxed supercoiling within the genome with organisms across all domains of life, not specifically prokaryotes. Maybe they are refereeing to negative supercoiling? (something not addressed in their work?) Without an extensive and believable explanation for what they mean I think this idea should be dropped. One idea to consider could be a link with DNA replication. DNA would be relaxed behind the DNA replication fork and could probably be suggested as a cue for initiating Figure-8 formation. This could allow replication factors to be more readily available to initiate replication of the second strand of each copy of the element and help displace the circle. If plasmids are expected to be supercoiled it could support the idea of favoring transposition to molecules that could facilitate horizontal transfer.

We thank the referee for the insights, and we agree that our invoking the global topological status of cellular DNA was not warranted; we now leave the issue restricted to our biochemical observations. We thank the referee for the suggestion to emphasize the possible role of replication as it is clear from models of this transposition pathway that replication must play a crucial role. We have modified the text accordingly:

"These observations suggest the intriguing possibility that transposition initiation and production of a Figure 8 intermediate might be restricted to the relaxed segments of bacterial genomic DNA that are found, for instance, associated with replication forks during DNA replication. This could also make host replication factors immediately available to process Figure 8 intermediates into their dsDNA circular form."

2. Optimal growth temperature for *Clostridium thermocellum* is 55C, but all of the experiments were done at 37C. Were normal growth temperatures for *C. thermocellum* tested *in vitro*? This could have some effect on the interpretation of supercoiling and AT bias and the complications with protein aggregation.

In our initial biochemical experiments, we conducted *in vitro* integration/circularization assays at 37 °C and 55 °C and the results were not dependent on the temperature. As our main focus was structural work and we did not observe any significant effects of temperature on purified protein stability, we performed all subsequent activity assays at 37 °C. We have added the following to p. 8:

"Although *C. thermocellum* is an anaerobic thermophile that grows optimally at 55 °C (Akinosho et al, 2014), we did not observe any effect on *in vitro* activity when the temperature was increased."

3. The paper just seems to fall off at the end without completing the thought at the end of the discussion?

As suggested, we have added a final summary paragraph to the discussion section.

4. In Figure 1B and 1C as shown, the donor and target DNAs are in opposite 5 prime to 3 prime orientations. Because each is turned on its side, it is hard to say which should be which, but for clarity the same convention should be used relative to each other.

We thank the referee for bringing this to our attention, and the orientations have been standardized.

5. Figure 1C would be better if the points of joining to the target were more clearly at the edges of what will become the TSD.

The referee is correct and the panel has been modified.

6. Figure 2A and D would be much clearer if it was shown with both DNA strands.

The figure panels have been modified to show both strands.

7. Figure 7 Why was the NDB domain left out of the "B" protomer? Was this because it lacked resolution in the structure, maybe show with dashed line instead?

Yes, the referee is correct that the absence of the NBD domain in protomer B in the scheme reflected our observation that this segment is unstructured in the complex. We agree that a dashed line is better, and the panel has been changed accordingly.

8. Page 3, paragraph 1 = "Many ISs also contain regulatory elements such as promoters..."

Clarify that talking about promoters that will transcribe the region outside the element upon integration.

We have revised the sentence as suggested:

"Many ISs also contain sequences that can act as promoters for genes located outside of the element and thus can dynamically affect their expression upon integration..."

9. Page 3, last line, paragraph 1 - Not all of these references given involve CRISPR-Cas9 and "efficiency" is vague. Should state something about guide-RNA targeting generally with CRISPR-Cas. Guide RNA targeting with Cas9 is a new function of the system so how can it be an improvement in "efficiency"?

Upon further reflection, we have deleted the sentence (and associated references) that refers to CRISPR-Cas9 from the revised manuscript as they are probably not relevant.

10. Page 4, last paragraph - "...which drives transposase expression..." should say something along the lines of "...drives increased transposase expression..." because the transposase must have been expressed at some level to allow excision of the element to make the circular form.

We thank the referee for pointing this out, which is of course correct, and we have modified the sentence as suggested.

11. Page 6, first paragraph - Write out *Clostridium thermocellum*.

This has been done.

Thank you for submitting your final revised manuscript for our consideration. I am pleased to inform you that in light of the positive re-review by referee 1 (copied below), we have now accepted it for publication in The EMBO Journal.

REFEREE REPORTS

Referee #1:

The authors have comprehensively addressed all the points raised in my initial review. The additional supplementary figure 7 is very useful and the main figures have been improved. I recommend acceptance.

Corresponding Author Name: Fred Dyda
Journal Submitted to: EMBO JOURNAL
Manuscript Number: EMBOJ-2020-105666